# Dual inhibition of the terminal oxidases eradicates antibiotic-tolerant *Mycobacterium tuberculosis*

Bei Shi Lee[1], Kiel Hards[2,3], Curtis A Engelhart[4], Erik J Hasenoehrl[5], Nitin P Kalia[6,7], Jared S Mackenzie[8], Ekaterina Sviriaeva[1], Shi Min Sherilyn Chong[1,9], Malathy Sony S Manimekalai[1], Vanessa H Koh[10,11], John Chan[12], Jiayong Xu[12], Sylvie Alonso[10,11], Marvin J Miller[13], Adrie J C Steyn[8,14], Gerhard Grüber[1], Dirk Schnappinger[4], Michael Berney[5], Gregory M Cook[2,3], Garrett C Moraski[15] & Kevin Pethe[1,6,*]

## Abstract

The approval of bedaquiline has placed energy metabolism in the limelight as an attractive target space for tuberculosis antibiotic development. While bedaquiline inhibits the mycobacterial $F_1F_0$ ATP synthase, small molecules targeting other components of the oxidative phosphorylation pathway have been identified. Of particular interest is Telacebec (Q203), a phase 2 drug candidate inhibitor of the cytochrome $bcc:aa_3$ terminal oxidase. A functional redundancy between the cytochrome $bcc:aa_3$ and the cytochrome $bd$ oxidase protects *M. tuberculosis* from Q203-induced death, highlighting the attractiveness of the $bd$-type terminal oxidase for drug development. Here, we employed a facile whole-cell screen approach to identify the cytochrome $bd$ inhibitor ND-011992. Although ND-011992 is ineffective on its own, it inhibits respiration and ATP homeostasis in combination with Q203. The drug combination was bactericidal against replicating and antibiotic-tolerant, non-replicating mycobacteria, and increased efficacy relative to that of a single drug in a mouse model. These findings suggest that a cytochrome $bd$ oxidase inhibitor will add value to a drug combination targeting oxidative phosphorylation for tuberculosis treatment.

**Keywords** antibiotic-tolerance; cytochrome $bcc$-$aa3$; cytochrome $bd$ oxidase; oxidative phosphorylation; Q203

**Subject Categories** Microbiology, Virology & Host Pathogen Interaction; Pharmacology & Drug Discovery

## Introduction

*Mycobacterium tuberculosis* is the causative agent of human tuberculosis (TB). Due to the threat TB poses to public health and global development, the World Health Organization (WHO) initiated the End TB strategy in 2014 (WHO, 2019). The initiative aims to reduce TB deaths by 95%, as well as to lower new cases by 90% by the year 2035. However, despite a global commitment to eradicate tuberculosis, the disease remains the foremost cause of death by a single infectious agent. Given the current trend of progress, we are predicted to fall short of the goal (WHO, 2019). The goal to rid the world of tuberculosis is further hindered by the emergence of multi-drug resistance to the current treatment. The traditional combination of isoniazid, rifampicin, pyrazinamide, and ethambutol is effective in clearing uncomplicated cases of tuberculosis; however, the emergence of resistance to rifampicin and isoniazid renders the traditional drug combination ineffective (WHO, 2019). While the anti-TB drug pipeline has begun to revive after a

1   School of Biological Sciences, Nanyang Technological University, Singapore, Singapore
2   Department of Microbiology and Immunology, School of Biomedical Sciences, University of Otago, Dunedin, New Zealand
3   Maurice Wilkins Centre for Molecular Biodiscovery, University of Auckland, Auckland, New Zealand
4   Department of Microbiology and Immunology, Weill Cornell Medical College, New York, NY, USA
5   Department of Microbiology and Immunology, Albert Einstein College of Medicine, Bronx, NY, USA
6   Lee Kong Chian School of Medicine, Nanyang Technological University, Singapore, Singapore
7   Ramalingaswami Fellow, Clinical Microbiology Division, CSIR-IIIM, Jammu and Kashmir, India
8   Africa Health Research Institute, Nelson R. Mandela School of Medicine, University of KwaZulu-Natal, Durban, South Africa
9   Nanyang Institute of Technology in Health and Medicine, Interdisciplinary Graduate School, Nanyang Technological University, Singapore, Singapore
10  Department of Microbiology, Yong Loo Lin School of Medicine, National University of Singapore, Singapore, Singapore
11  Infectious Disease Programme, Department of Microbiology and Immunology, National University of Singapore, Singapore, Singapore
12  Department of Medicine, Albert Einstein College of Medicine, Bronx, NY, USA
13  Department of Chemistry and Biochemistry, University of Notre Dame, Notre Dame, IN, USA
14  Department of Microbiology, University of Alabama, Birmingham, AL, USA
15  Department of Chemistry and Biochemistry, Montana State University, Bozeman, MT, USA
*Corresponding author. Tel: +65 65923967; E-mail: kevin.pethe@ntu.edu.sg

40-year drought—with bedaquiline, delamanid, and pretomanid approved for use in the treatment against multi- and extensively drug-resistant TB—the rapid emergence of resistance to these newly approved drugs is a major source of concern (Bloemberg et al, 2015). Thus far, MDR TB still has an unfavorable prognosis (56% treatment success rate (WHO, 2019)), and its rising incidence is rightfully becoming a source of concern for many countries. Novel drugs effective against MDR- and XDR-TB are thus in urgent demand.

*Mycobacterium tuberculosis* infections are notoriously difficult to treat due to several reasons. Of which, one of the most prominent is the long treatment time, often lasting at least four to six months on a four-drug cocktail in uncomplicated cases (WHO, 2019). This uniquely long therapy is believed to be due to a persistent subpopulation of bacteria which is able to survive for extended periods of time without growth, and is tolerant to the first-line drugs (Dick, 2001; Wayne & Sohaskey, 2001; Betts et al, 2002; Boshoff & Barry, 2005). *Mycobacterium tuberculosis* can enter a non-replicating, antibiotic-tolerant state upon exposure to certain environmental stresses such as low oxygen tension or nutrient availability (Wayne & Sohaskey, 2001; Boshoff & Barry, 2005). The ability for the bacilli to survive in oxygen-poor environments found in granulomas is widely reported (Wayne & Sohaskey, 2001; Ramakrishnan, 2012), and simulations of hypoxia *in vitro* to study their adaptations were developed (Wayne & Hayes, 1996). Nutrient scarcity has also been used as a stress factor to induce a persistence state (Betts et al, 2002). It is important to note that this quiescence is not permanent, and the bacteria have the capacity to emerge from their persistent state when conditions are more favorable (Wayne & Sohaskey, 2001; Boshoff & Barry, 2005).

Hence, with the aim of shortening TB treatment, it is only logical to pursue the development of novel drugs that are active against antibiotic-tolerant, non-replicating *Mycobacterium tuberculosis*. Although the physiology of non-replicating mycobacteria is not fully understood, bioenergetics is a validated target space (Koul et al, 2008; Rao et al, 2008; Gengenbacher et al, 2010; Berube et al, 2019). Membrane bioenergetics governs the maintenance of the proton motive force and ATP homeostasis. The most prominent inhibitor of *M. tuberculosis* energy metabolism is the $F_1F_0$-ATP synthase inhibitor, bedaquiline (Andries et al, 2005; Elias et al, 2013), that was approved for clinical use in 2012 (Mahajan, 2013). Its bactericidal potency against non-replicating subpopulations and high efficacy in man validated energy metabolism as an attractive target space for drug development (Koul et al, 2008; Hards et al, 2015). Following the development of bedaquiline, a great deal of focus turned to the exploration of energy metabolism for additional vulnerable targets. Several small-molecule drugs acting on the oxidative phosphorylation pathway have been identified. Several preclinical-stage drugs targeting the type II NADH dehydrogenases (Weinstein et al, 2005; Warman et al, 2013; Dunn et al, 2014), the cytochrome $bcc:aa_3$ (Moraski et al, 2011; Abrahams et al, 2012; Moraski et al, 2013; Pethe et al, 2013; Rybniker et al, 2015; Moraski et al, 2016), and menaquinone synthesis were discovered (Lu et al, 2008; Debnath et al, 2012; Lu et al, 2012). Telacebec (Q203), a drug candidate targeting the cytochrome $bcc:aa_3$ terminal oxidase (Pethe et al, 2013), recently demonstrated a favorable safety profile and potency in a phase 2 clinical trial (de

Jager et al, 2020). Q203 binds to the cytochrome b subunit of the cytochrome bcc and is bacteriostatic in low nanomolar concentration (Pethe et al, 2013). Q203's sterilizing potency is limited by the presence of the cytochrome bd oxidase, an alternative terminal oxidase (Kalia et al, 2017). The synergistic lethality between the cytochrome $bcc:aa_3$ (Cyt-$bcc:aa_3$) and the cytochrome bd oxidase (Cyt-bd) represents an untapped therapeutic potential to develop a rational, sterilizing drug combination for tuberculosis (Arora et al, 2014; Kalia et al, 2017; Moosa et al, 2017). The only known Cyt-bd inhibitor is the non-selective menaquinone analogue Aurachin D (Lu et al, 2018). However, the toxicity of the compound coupled with its lack of selectivity and lipophilic nature precludes further chemical optimization (Dejon & Speicher, 2013; Li et al, 2013; Schäberle et al, 2014).

Here, we report the discovery and characterization of a small molecule targeting the Cyt-bd that together with Q203 forms a bactericidal drug combination against replicating *M. tuberculosis*, as well as hypoxic, and nutrient-starved antibiotic-tolerant subpopulations.

## Results

### Identification of the putative Cyt-bd inhibitor ND-011992

We developed a facile whole-cell screen to identify Cyt-bd inhibitors. The assay principle relies on the conditional essentiality of the Cyt-bd to maintain ATP homeostasis when the function of the Cyt-$bcc:aa_3$ is pharmacologically inhibited. Q203 on its own lowers intracellular ATP levels in *M. tuberculosis* and *Mycobacterium bovis* BCG, albeit to a lesser degree compared with the effect triggered by bedaquiline (Kalia et al, 2017). For practical biosafety reasons, the assay was initially developed in *M. bovis* BCG. *M. bovis* BCG is an excellent surrogate mycobacteria to study the terminal oxidases since its Cyt-$bcc:aa_3$ and Cyt-bd share 100% sequence similarity with the *M. tuberculosis* H37Rv counterparts (Brosch et al, 2007; Lew et al, 2011; Data ref: Garnier, 2006; Data ref: Lew, 2012). Screening of a collection of 53 small molecules in the presence or absence of Q203 led to the identification of two quinazolin-4-amine small molecules. The small molecules were selected from a larger library assembled during an NIH project focused on the design and discovery of novel small molecules for tuberculosis treatment (Moraski et al, 2012; Tiwari et al, 2013; Moraski et al, 2014). The representative molecule ND-011992 (Fig 1A) depleted ATP levels only in the presence of Q203 at an inhibitory concentration 50% ($IC_{50}$) of 0.5–1.6 μM in *M. bovis* BCG (Fig 1B). The compound was also active against *M. tuberculosis* H37Rv at an $IC_{50}$ of 2.8–4.2 μM (Fig EV1A). An adapted checkerboard assay demonstrated a synergistic ATP depletion triggered by the combination of Q203 and ND-011992, achieving a minimum fractional inhibitory concentration (FIC) index of 0.01 in *M. bovis* BCG and 0.16 in *M. tuberculosis* (Fig EV1B and C), reflecting a high degree of positive interaction. Furthermore, ND-011992 lowered the minimum inhibitory concentration 50% ($MIC_{50}$) of Q203 from 3.16 to 0.97 nM in *M. tuberculosis* H37Rv (Fig EV1D). The higher $IC_{50}$ of ND-011992 in *M. tuberculosis* H37Rv may be explained by a high basal expression of the Cyt-bd-encoding genes in laboratory-adapted *M. tuberculosis* strains (Arora et al, 2014).

### ND-011992 inhibits oxygen consumption in the presence of Q203

The potency of ND-011992 was next tested for its ability to inhibit oxygen consumption. Methylene blue was first used as a qualitative indicator of dissolved oxygen level. The combination of Q203 and ND-011992 arrested oxygen consumption in *M. bovis* BCG (Fig 1C) and *M. tuberculosis* H37Rv (Fig EV2A), while each individual drug alone had no apparent effect. The phenotype was confirmed by quantitative microplate-based respirometry. The combination of Q203 and ND-011992 resulted in an immediate and complete inhibition of the oxygen consumption rate (OCR) of *M. bovis* BCG on a Seahorse XFe96 platform (Fig 1D). In combination with a fixed dose of Q203, ND-011992 inhibited the mycobacterial OCR at an $IC_{50}$ of 0.8 μM (Fig 1E). The drastic inhibitory effect of the combination Q203 + ND-011992 was confirmed using the MitoXpress® Xtra oxygen consumption assay in both *M. bovis* BCG and *M. tuberculosis*

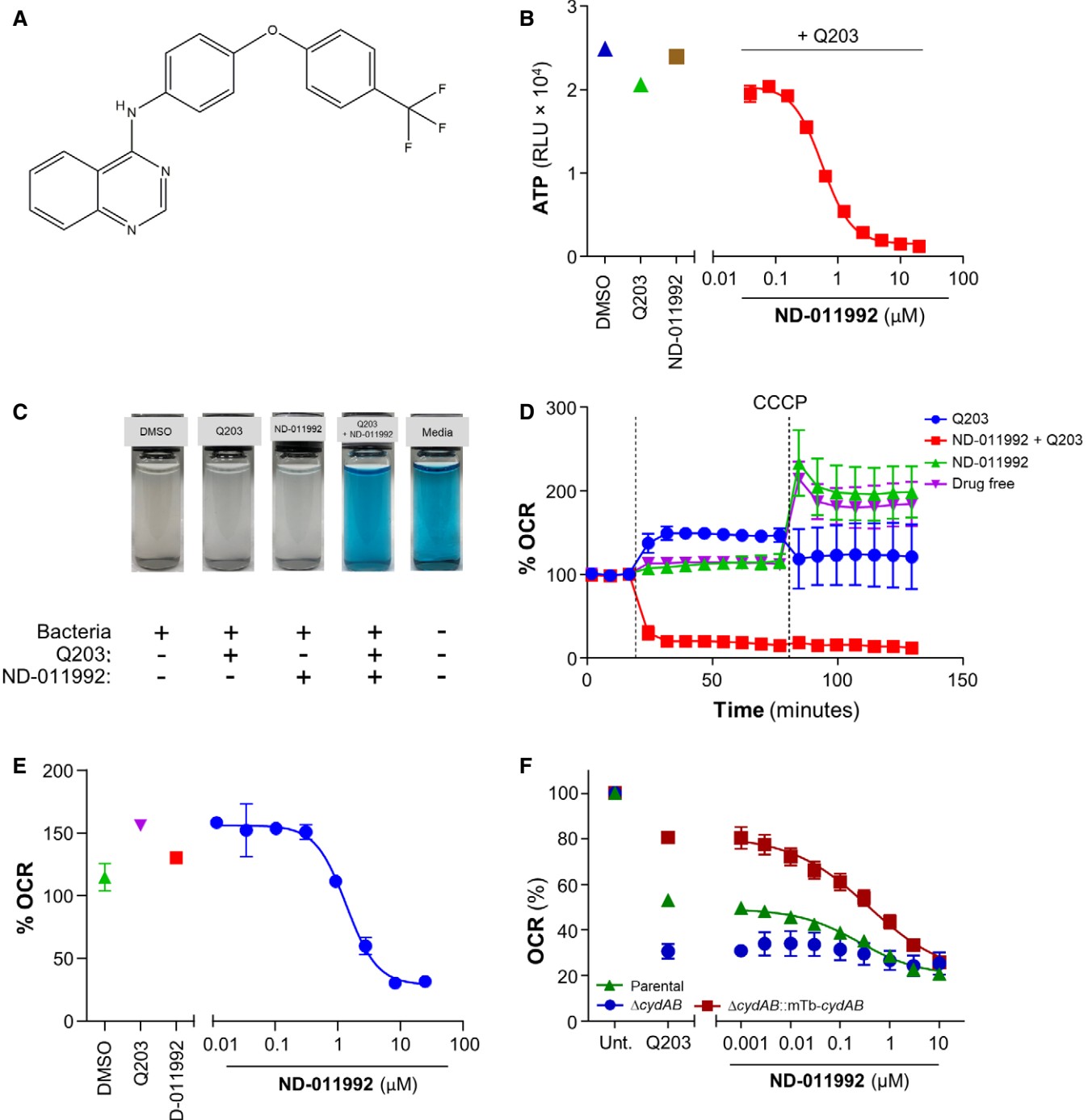

**Figure 1.**

**Figure 1.   Identification of the putative Cyt-*bd* inhibitor ND-011992.**

A   Molecular structure of ND-011992.

B   Effect of ND-011992 on the intracellular ATP level in *M. bovis* BCG. The bacteria were treated with DMSO (blue triangle), 100 nM Q203 alone (green triangle), 20 μM ND-011992 alone (brown square), and a dose range of ND-011992 in the presence of Q203 (red squares) for 15 hours before quantification of intracellular ATP levels.

C   Oxygen consumption assay in *M. bovis* BCG using methylene blue as an oxygen sensor.

D   Oxygen consumption assay in *M. bovis* BCG using the Seahorse XFe96 Extracellular flux analyzer. 12 μM ND-011992 was injected alone (green triangles), 100 nM Q203 (blue circles), or in combination with Q203 (red squares). For each condition, OCR readings were normalized to the last basal OCR reading before drug injection. OCR: oxygen consumption rate. The dotted lines indicate the timepoints of drug injection.

E   Dose-dependent inhibition of *M. bovis* BCG OCR by ND-011992 (in the presence of 100 nM Q203) measured on a Seahorse XFe96 analyzer platform. For each condition, OCR readings were normalized to the last basal OCR reading before drug injection.

F   Effect of ND-011992 on the OCR of *M. smegmatis* IMVs using the Oroboros O$_2$k fluorespirometer. IMV OCR from the parental strain (green triangles), Δ*cydAB* knockout (blue circles), and Δ*cydAB* complemented with *M. tuberculosis* CydABDC$^+$ (red squares) energized with NADH were determined. Q203 was used at 1 μM. 100% OCR was defined as the OCR of the untreated samples for each strain.

Data information: Data are expressed as the mean ± SD for each condition of a representative experiment. Experiments in (B) and (E) were performed with 2 technical replicates and independently repeated at least twice. In (D) and (F), each experiment was performed with 3 technical replicates and independently repeated at least once.

H37Rv (Fig EV2B–E). Biochemical assays were performed on purified inverted-membrane vesicles (IMVs) from *Mycobacterium smegmatis* mc²155 to obtain preliminary evidence that ND-011992 acts on the Cyt-*bd* in the lipid-rich environment of the IMVs. IMVs purified from *M. smegmatis* were used since protocols for their purification in high amount and subsequent characterization of the associated electron transport chain are well-established (Hards *et al*, 2015; Heikal *et al*, 2016; Lu *et al*, 2015; Pecsi *et al*, 2014). To provide further evidence of target engagement of ND-011992 with the *M. tuberculosis* Cyt-*bd* in a lipid-rich environment, we prepared IMVs from an *M. smegmatis* Δ*cydAB* mutant expressing the *M. tuberculosis cydABDC* operon. Results revealed that a combination of Q203 and ND-011992 inhibited oxygen consumption, whereas each drug individually had limited effects, a finding consistent with the behavior of a direct Cyt-*bd* inhibitor (Fig 1F). The use of IMVs may enhance the potency of Q203 as the target QcrB is more readily accessible to chemical inhibition by Q203 and hence amplifying the effect of OCR inhibition in these experiments. Importantly, the OCR of IMVs prepared from a *M. smegmatis* Δ*cydAB* mutant was inhibited by Q203, but not with increasing concentrations of ND-011992. When the Δ*cydAB* mutant strain was complemented with the *cydABDC* operon from *M. tuberculosis*, OCR sensitivity to ND-011992 was restored, supporting that the cytochrome *bd* oxidase is the primary target of ND-011992 (Fig 1F). Additional experiments conducted on IMVs purified from the attenuated strain *M. tuberculosis* mc²6230 confirmed that the drug combination inhibited the electron transport chain in *M. tuberculosis* (Fig EV2F).

## Mode of action studies of ND-011992

The transcriptional responses of *M. tuberculosis* H37Rv to Q203, ND-011992, and a combination of both were measured by RNA-seq (Fig 2A). H37Rv treated with ND-011992 alone displayed only 5 statistically significant upregulated or downregulated genes. Among highly upregulated genes were *ethA2* (*Rv0077c*), which is part of a cryptic drug-bioactivation pathway that has been proposed to circumvent ethionamide resistance in the cell (Blondiaux *et al*, 2017). However, ND-011992 did not lower the MIC of ethionamide in *M. tuberculosis* H37Rv (Appendix Fig S1). Treatment with Q203 in H37Rv induced significant changes in 675 genes, and the most

affected genes are involved in the respiratory chain and redox stress, which is a typical signature for drugs targeting respiration (Boshoff *et al*, 2004; Fig 2A). The greatest number and magnitude of gene expression changes were seen in H37Rv treated with the combination of Q203/ND-011992 (1,147 genes) and with the Δ*cydAB* strain treated with Q203 alone (1,369 genes). The majority of these genes were downregulated, and of the 891 downregulated genes in the combination-treated H37Rv strain, more than two-thirds (67.6%) were similarly downregulated in the Δ*cydAB* mutant challenged with Q203 alone. Interestingly, the transcription of respiratory chain and redox stress response genes was particularly affected by the addition of ND-011992, indicating a more drastic poisoning of the electron transport chain (Fig 2A). Since the transcriptional response of H37Rv treated with the Q203/ND-011992 combination was almost identical to the response of H37Rv Δ*cydAB* treated with Q203 alone, it can be inferred that the transcriptomic response induced by ND-011992 is compatible with a specific Cyt-*bd* inhibition.

Next, a series of experimental approaches were carried out to validate target engagement further. Increasing the gene copy number of the cyd*ABDC* operon from a high-copy number plasmid caused an 18- to 31-fold shift in ND-011992 potency from 0.52 to 17.2 μM, as well as a 12-fold shift in OCR from 0.62 to 7.8 μM (Fig 2B). The potencies of Q203 and bedaquiline were unaffected in the Cyt-*bd* overexpressing strain, indicating that ND-011992 inhibits Cyt-*bd* specifically (Appendix Table S1). Furthermore, given the functional redundancy between both terminal oxidases, we reasoned that ND-011992 alone should be a more potent mycobacterial growth inhibitor in the absence of the Cyt-*bcc:aa*$_3$. Consistently, ND-011992 inhibited the growth of an H37Rv strain deficient in Cyt-*bcc: aa*$_3$ expression (Beites *et al*, 2019) at a concentration 33- to 110-fold lower compared with the concentration required to inhibit the growth of the parental strain (Fig 2C and Appendix Table S2).

Target engagement was further demonstrated on mycobacterial inverted-membrane vesicles. The difference spectra of IMVs from the parental *M. smegmatis* mc²155 and the *M. smegmatis* Δ*qcrCAB* mutant—with the latter expressing a higher Cyt-*bd* level—were employed to identify which cytochrome oxidase hemes were affected by ND-011992 (Kana *et al*, 2001; Megehee *et al*, 2006; Gong *et al*, 2018; Hards *et al*, 2019). The difference spectrum of the

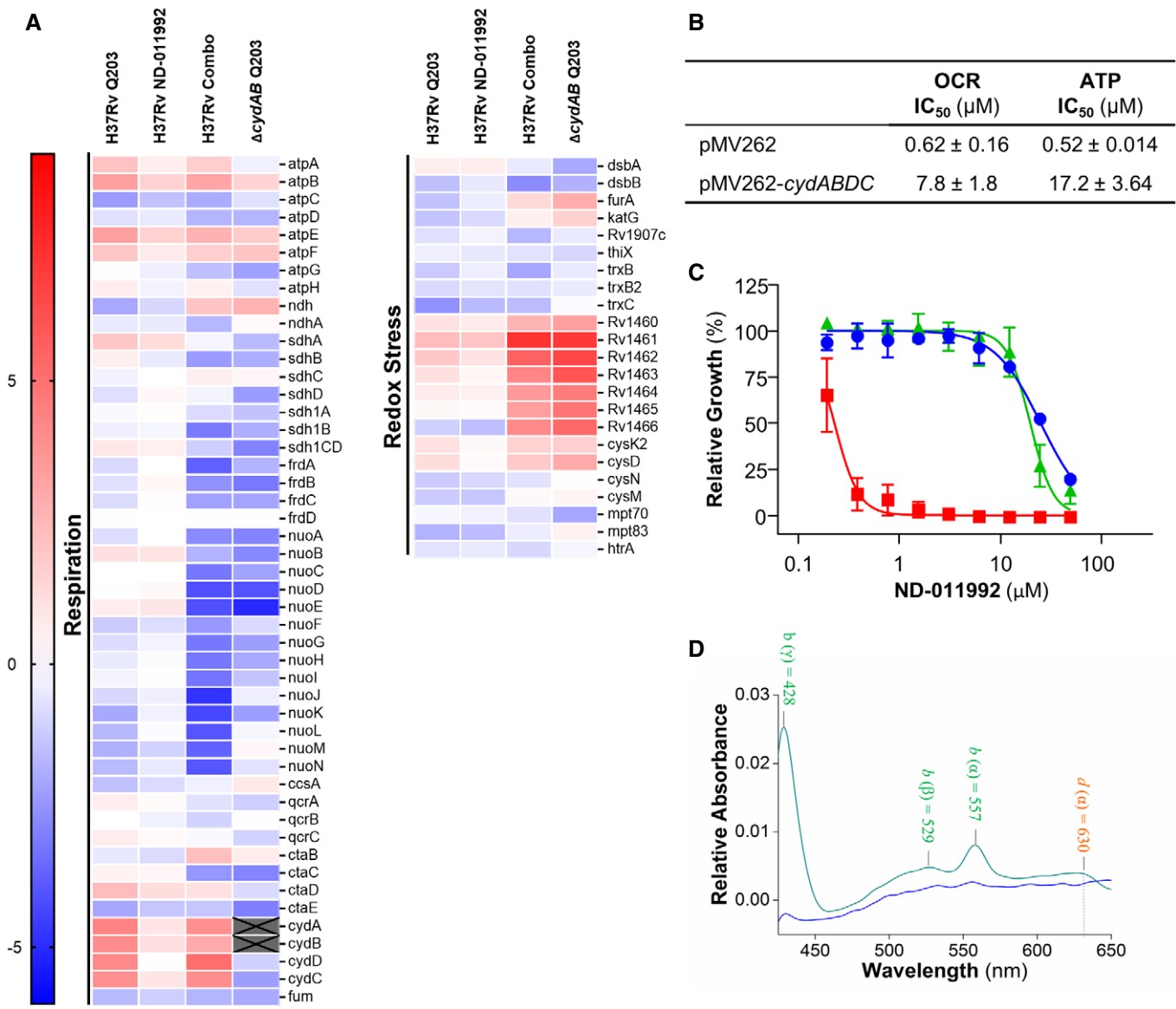

**Figure 2. ND-011992 inhibits the mycobacterial Cyt-*bd*.**

A Differential gene expression analysis of H37Rv treated with Q203, ND-011992, or combination (H37Rv-Combo), and Δ*cydAB* treated with Q203 compared with untreated controls. The log$_2$-fold differences in gene expression of various conditions relative to the untreated control are indicated using a sliding scale where higher expression is reflected in red and lower expression is reflected in blue, with a midpoint signifying no difference in white.

B Effect of *cydABDC* overexpression on the oxygen consumption rate (OCR) and intracellular ATP levels. In *M. bovis* BCG treated with ND-011992 in the presence of 100 nM Q203. IC$_{50}$: inhibitory concentration 50%.

C Absence of the Cyt-*bcc:aa$_3$* sensitized H37Rv to ND-011992. *M. tuberculosis* H37Rv wild type (blue circles), Δ*ctaE-qcrCAB* (red squares), and complemented strain (green triangles) were treated with ND-011992. Growth inhibitory potency (MIC$_{50}$) was recorded after 2 to 3 weeks of incubation.

D The reduced minus oxidized difference spectrum of *M. smegmatis* Δ*qcrCAB* inverted-membrane vesicles (IMVs). The green line represents the spectrum without treatment, while the blue line represents the spectrum of the IMVs in the presence of 10 μM ND-011992.

Data information: Data are expressed as the mean ± SD of triplicates for each condition of a representative experiment. Each experiment was performed with 3 technical replicates and independently repeated at least once.

Source data are available online for this figure.

*M. smegmatis* Δ*qcrCAB* mutant IMVs energized with NADH (2 mM final concentration) demonstrated electron transfer followed by the reduction in the cytochrome *bd* hemes (*b* at 428, 529, and 557 nm, and *d* at 630 nm) (Fig EV3A). Similarly, NADH-driven reduction in

the cytochrome *bcc:aa$_3$* oxidase hemes (*a* at 443 and 600 nm, *b* at 432 and 563 nm, and *c* at 523 and 552 nm) was also clearly assigned for the parental *M. smegmatis* mc$^2$155 IMVs (Fig EV3A). In comparison, the difference spectrum of the parental *M. smegmatis*

mc$^2$155 IMVs in the presence of NADH and the cytochrome *bcc* inhibitor Q203 displayed a significant change in the peaks corresponding to the hemes of the cytochrome *bcc-aa₃* supercomplex (Fig EV3B), confirming drug binding and inhibition of the electron transfer within the cytochrome *bcc* complex (Pethe *et al*, 2013; Kalia *et al*, 2017; Bouvier *et al*, 2019). When the same experiment was performed in the presence of ND-011992, no major alterations of the profiles of the heme peaks of the Cyt-*bcc:aa₃* supercomplex were detected (Fig EV3C). Importantly, when the difference spectrum of the *M. smegmatis* Δ*qcrCAB* IMVs energized with NADH and treated with ND-011992 was collected, the peaks corresponding to the Cyt-*bd* hemes were strongly reduced or disappeared (Fig 2D). This result demonstrated that the compound selectively binds and affects electron transfer in the mycobacterial Cyt-*bd,* indicating that ND-011992 is a quinone binding site inhibitor.

### ND-011992 is active across phylogenetic lineages and has a low rate of spontaneous resistance

Potency of a new drug candidate against *M. tuberculosis* isolates other than a laboratory-adapted strain is a requirement. The efficacy of ND-011992 was evaluated against a panel of well-characterized pan-susceptible isolates (Gagneux, 2018; Borrell *et al*, 2019), and four drug-resistant clinical isolates in an agar plate MIC assay. The assay principle relies on the observation that Q203 is less than bacteriostatic against *M. tuberculosis* strains on glycerol-supplemented agar plates due to the presence of the Cyt-*bd* (Kalia *et al*, 2019). ND-011992 was potent at an MIC value ranging from ≤ 0.2 to 1 μM against pan-susceptible clinical isolates, and MIC ≤ 1 μM against the four multi- and extensively drug-resistant clinical isolates of South African origin (Fig 3A and B), showing that the drug is active against a diverse pool of clinical isolates, including those resistant to first- and second-line antituberculosis agents.

Antibiotic synergistic interactions may impose an increased pressure to select for spontaneous resistance (Hegreness *et al*, 2008). The frequency of spontaneous resistance to a combination of ND-011992 and Q203 was determined. When mutant selection was attempted on either Q203 alone or ND-011992 alone, a lawn was obtained after 25 days of incubation, reflecting the time-dependent loss of potency of Q203 on glycerol-supplemented medium as reported before (Kalia *et al*, 2019), and the lack of growth inhibitory potency of ND-011992. When both drugs were tested in combination, the plates remained clear (Fig 3C), with few colonies occurring at a frequency of $6.6 \times 10^{-9}$ and $2.1 \times 10^{-9}$ in *M. bovis* BCG and *M. tuberculosis*, respectively, values which were one order of magnitude lower than Q203 alone (Appendix Table S3). Given that the Cyt-*bcc:aa₃* and Cyt-*bd* of *M. bovis* BCG and *M. tuberculosis* H37Rv share 100% sequence similarity, escape mutants isolated from *M. bovis* BCG were selected as representatives in the subsequent studies. All 18 *M. bovis* BCG colonies analyzed were highly resistant to Q203 (MIC$_{50}$ > 100 nM) (Table 1), suggesting that the primary mechanism of resistance to the drug combination is mediated by a mutation in QcrB (Pethe *et al*, 2013). To exclude that any of the escape mutants were co-resistant to ND-011992, we made use of lansoprazole, an inhibitor of the mycobacterial cytochrome *bcc:aa₃* that retains efficacy against Q203-resistant spontaneous mutants (Rybniker *et al*, 2015). Upon confirmation that lansoprazole retained efficacy against the spontaneous-resistant mutants (Table 1), we tested the inhibitory activity of a dose

range of ND-011992 in the same intracellular ATP assay used in the initial screen (Fig 1B), but in the presence of lansoprazole instead of Q203. ND-011992 depleted intracellular ATP levels at IC$_{50}$ values comparable to those observed in the parental *M. bovis* BCG strain (Table 1), indicating that the escape mutants remained fully susceptible to ND-011992. Despite repeated attempts, escape mutants resistant to ND-011992 could not be isolated, suggesting a frequency of spontaneous resistance below $3.7 \times 10^{-10}$. Fluctuation analysis was conducted to confirm the low propensity of resistance acquisition to the drug combination. In this experiment, we used the H37Rv Δ*cydAB* strain to compare the frequency of resistance to Q203 + ND-011992 to a reference. The H37Rv parental strain could not be used due to the bacteriostatic nature of Q203 and the loss of potency of the drug on agar plates supplemented with glycerol (Kalia *et al*, 2019). In addition, rifampicin was added as an additional control to compare the frequency of resistance to Q203 + ND-011992 to an approved drug. Fluctuation analysis results confirmed the finding that addition of ND-011992 to Q203 did not raise the pathogen's spontaneous frequency of resistance to the treatment (Fig 3D).

### ND-011992-Q203 combination is bactericidal in *M. tuberculosis* and potent *in vivo*

The main limitation of Q203 and related QcrB inhibitors lies in their lack of bactericidal efficacy (Kalia *et al*, 2017; Foo *et al*, 2018). In line with the synergistic lethal interaction between the two terminal oxidases, a combination of Q203 and ND-011992 was bactericidal against *M. tuberculosis* (Fig 4A) and *M. bovis* BCG (Fig EV4A). Furthermore, the drug combination was efficacious against both antibiotic-tolerant nutrient-starved (Fig 4B) and hypoxic non-replicating mycobacteria (Figs 4C and EV4B). Regardless of the metabolic state, the drug treatment achieved at least 95% killing efficacy at comparable drug concentrations (Fig 4A-C). The lack of differential tolerance levels to this treatment is indicative of the targets' essentiality in all three *in vitro* models, which is remarkable since most antibiotics are much less potent against non-replicating subpopulations (Nathan & Barry, 2015). The fact that chemical inhibition of both terminal oxidases kills *M. tuberculosis* irrespective of its metabolic and growth status emphasizes the attractiveness of the respiration pathway for anti-TB drug development.

We next determined the suitability of ND-011992 as a starting point for chemical optimization. The drug was stable in mouse plasma, in murine and human microsomes, as well as in simulated gastric fluid (Appendix Table S4). Despite a low solubility and low permeability in a Caco-2 assay, ND-011992 had a bioavailability of 58% in mice, a moderate volume of distribution (5.35 l/kg body weight) and very low systemic clearance (0.96 ml/min/kg), resulting in a prolonged half-life of 64 h (Appendix Table S5). The maximum concentration in serum was 1.6 μg/ml after a single oral dose of 10 mg/kg, which is moderate in relation to the drug potency (Appendix Fig S2). Despite a limited drug exposition and low cell permeability, we evaluated the efficacy of ND-011992 in combination with Q203 in an acute mouse model of tuberculosis. Drug treatment was initiated five days after intranasal infection with a high mycobacterial dose. The animals received 5 doses of either ND-011992 alone, Q203 alone, or a combination of both. Bacterial loads in the lung (Fig 4D) and spleen (Appendix Fig S3) from animals treated with the drug combination were significantly reduced

**A**

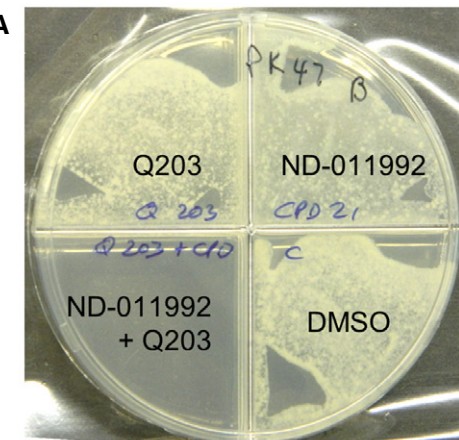

**C**

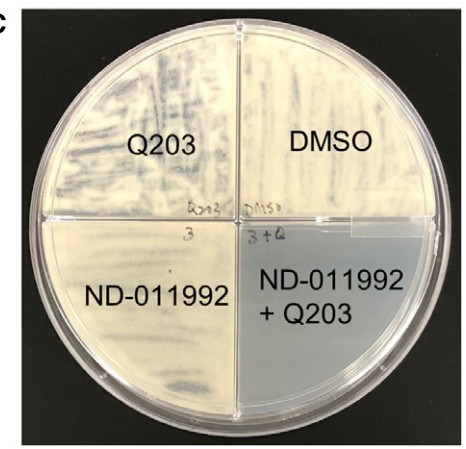

**B**

| Strain ID | Profile | Minimum inhibitory Concentration |
|---|---|---|
| | | Q203 + ND-011992 (µM) |
| H37Rv | S | 2.0 |
| N0072 | S | ≤ 0.2 |
| N0157 | S | ≤ 0.2 |
| N0145 | S | ≤ 0.2 |
| N0052 | S | 0.5 |
| N0054 | S | 0.2 |
| N1274 | S | 1.0 |
| N1283 | S | ≤ 0.2 |
| N0136 | S | 1.0 |
| 123-20-0015 | XDR | ≤ 1.0 |
| 123-20-0091 | XDR | ≤ 1.0 |
| 123-20-0041 | MDR | ≤ 1.0 |
| 123-20-0047 | MDR | ≤ 1.0 |

**D**

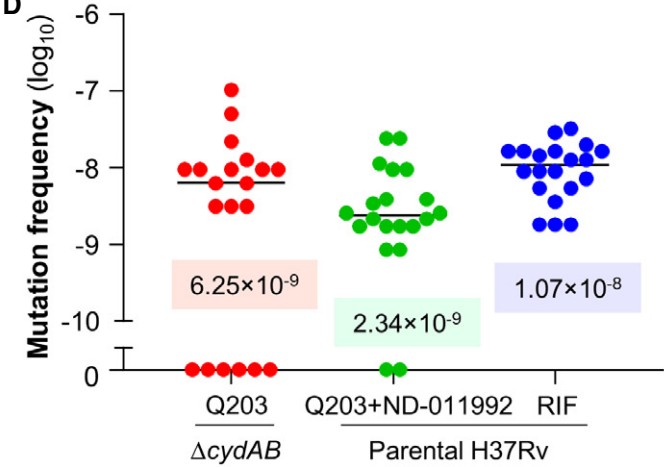

**Figure 3. ND-011992 has a low spontaneous resistance mutation frequency and is active against drug-resistant *M. tuberculosis* clinical isolates.**

A  Combined potency of ND-011992 + Q203 against the drug-resistant clinical isolate *M. tuberculosis* 123-20-0047.

B  Potency of the combination ND-011992 + Q203 against various clinical isolates in an agar plate MIC assay. A dose range of ND-011992 was tested against the clinical isolates in the presence of Q203. Q203 was used at 100 nM against the strains H37Rv, N1274, N1283, 123-20-0015, 123-20-0091, 123-20-0041, 123-20-0047, and at 20 nM against the stains N0072, N0157, N0145, N0052, and N0136. "S" indicates pan-susceptibility to anti-TB drugs; "MDR": multi-drug resistant; "XDR": extensively drug-resistant.

C  Growth inhibition assay in *M. bovis* BCG. $10^6$ bacteria were plated onto 7H10 agar supplemented with DMSO, 25 nM Q203, 3 µM ND-011992, or 25 nM Q203 + 3 µM ND-011992.

D  Fluctuation analysis in *M. tuberculosis* H37Rv. *M. tuberculosis* ΔcydAB was plated on 7H10 containing 100 nM Q203 (red). Parental H37Rv was plated on 7H10 with 100 nM Q203 and 6 µM ND-011992 or 2 µg/ml rifampicin (blue). The corresponding values of the median frequency of resistance are indicated in the graph.

**Table 1. Escape mutants selected on ND-011992 + Q203 are resistant to Q203 but not to ND-011992[†]**

| Strain | MIC$_{50}$ | | ND-011992 ATP IC$_{50}$ ($\mu$M) |
|---|---|---|---|
| | Q203 (nM) | Lansoprazole ($\mu$M) | |
| Wild type | 0.0014 | 44.61 | 0.74 |
| 1 | >100 | 55.94 | 0.98 |
| 2 | >100 | 57.46 | 0.92 |
| 3 | >100 | 54.19 | 0.90 |
| 4 | >100 | 57.5 | 0.48 |
| 5 | >100 | 56.06 | 0.53 |
| 6 | >100 | 58.22 | 0.76 |
| 7 | >100 | 55.71 | 0.50 |
| 8 | >100 | 63.5 | 0.40 |
| 9 | >100 | 60.94 | 0.58 |
| 10 | >100 | 52.37 | 0.83 |
| 11 | >100 | 99.5 | 0.98 |
| 12 | >100 | 68.33 | 0.77 |
| 13 | >100 | 62.92 | 0.67 |
| 14 | >100 | 63.62 | 0.87 |
| 15 | >100 | 63.88 | 0.56 |
| 16 | >100 | 64.38 | 0.38 |
| 17 | >100 | 63.79 | 0.60 |
| 18 | >100 | 62.87 | 0.46 |

The *M. bovis* BCG spontaneous-resistant mutant strains were exposed to a dose range of Q203 or lansoprazole for 5 days before recording the turbidity at OD$_{600}$. The potency of ND-011992 was tested in an ATP depletion assay in combination with lansoprazole. Bacteria were exposed to a dose range of ND-011992 in the presence of lansoprazole at 200 $\mu$M for 15 h before quantification of intracellular ATP levels. The experiments were performed in duplicates and repeated once.

compared with Q203, whereas ND-011992 had no potency. These results suggest that the concept is translatable into a mouse model of tuberculosis infection and that an optimized version of ND-011992 with improved pharmacokinetic properties may synergize further with Q203 or related QcrB inhibitors *in vivo*.

# Discussion

Two challenges associated with TB are the rapid spread of MDR and XDR cases (WHO, 2019), and the lack of drug combinations that can rapidly eradicate infection. The latter challenge is attributed to the presence of antibiotic-tolerant subpopulations (Dick, 2001; Wayne & Sohaskey, 2001; Betts *et al*, 2002; Boshoff & Barry, 2005; Zhang *et al*, 2012) that can survive in the absence of growth for extended periods of time. We and others have established that oxidative phosphorylation is essential in the maintenance of bioenergetics homeostasis in antibiotic-tolerant hypoxic (Rao *et al*, 2008) and nutrient-starved (Gengenbacher *et al*, 2010) *M. tuberculosis*,

opening a scientific rationale to eradicate antibiotic-tolerant, non-replicating subpopulations (Cook *et al*, 2014; Hards & Cook, 2018; Kalia *et al*, 2017; Koul *et al*, 2008). *M. tuberculosis* being an obligate aerobe and has two aerobic respiratory branches that are jointly essential for growth and viability (Kalia *et al*, 2017; Beites *et al*, 2019). Between the two terminal oxidases, the Cyt-*bcc:aa₃* is particularly vulnerable to chemical inhibition (Moraski *et al*, 2011; Abrahams *et al*, 2012; Moraski *et al*, 2013; Pethe *et al*, 2013; Rybniker *et al*, 2015; Moraski *et al*, 2016). The recent demonstration that the Cyt-*bcc:aa₃* inhibitor Q203 is potent in the two-week early bactericidal activity (EBA) proof-of-concept study in man provided another promising drug candidate for TB treatment (de Jager *et al*, 2020). However, we need to be cognizant that the potency of Q203 observed in an EBA study may not translate into treatment shortening unless the drug candidate is combined with appropriate companion drugs. For instance, it is well reported that the functional redundancy between the Cyt-*bcc:aa₃* and the Cyt-*bd* makes Q203 bacteriostatic and inactive against antibiotic-tolerant populations (Arora *et al*, 2014; Kalia *et al*, 2017; Moosa *et al*, 2017). While the prospect is alluring, identification of chemically tractable Cyt-*bd* inhibitors is not straightforward given the target's non-strict essentiality for growth under normal conditions. In this study, we exploited the ability of the Cyt-*bd* to partially maintain ATP homeostasis when the function of the Cyt-*bcc:aa₃* is inhibited by Q203 to identify putative hit candidates. The assay is rapid (15 h) and could be miniaturized to screen large compound collections. Our chemical screen led to the identification of the quinazolin-4-amine Cyt-*bd* inhibitor named ND-011992. In both qualitative and quantitative measurements of oxygen consumption, ND-011992 alone had negligible effect on oxygen consumption rates, while its combination with Q203 abruptly halted the bacilli's oxygen consumption. Consistent results yielded from membrane vesicles furthered our confidence in ND-011992's binding target as IMVs offer a closer examination of the compounds' effects on the membrane-bound ETC, while other possible confounding factors associated with whole cells are greatly reduced. Furthermore, analysis from transcriptomics revealed similar gene expression signature between parental *M. tuberculosis* treated with the ND-011992-Q203 combination and Δ*cydAB M. tuberculosis* treated with Q203. This observation drew a parallel between ND-011992 inhibition and Cyt-*bd* genetic knockout, and provided additional evidence that ND-011992 binds and inhibits the Cyt-*bd*. From the testing of pan-susceptible isolates across different lineages, it was demonstrated that the various isolates were all susceptible to chemical inhibition of the two terminal oxidases, albeit with slight variations in the degree of vulnerability. The reason behind the disparity is unclear but could arise from differences in expression levels of the two terminal oxidases. Nevertheless, the clear potency of ND-011992 against different lineages, regardless of their antibiotic-resistance profile, shows that the compound possesses a key attribute for further development. As with any therapeutics, the likelihood of resistance emergence might foretell the prospects of successful treatment outcome and is to be examined closely during development. Frequency of resistance (FOR) measures the incidences of spontaneous mutations that confer drug resistance. Since chemical inhibition of both terminal oxidases is strictly needed to achieve effective killing, mutations on

[†]Correction added on 11 January 2021, after first online publication: the concentration of Q203 was changed from $\mu$M to nM.

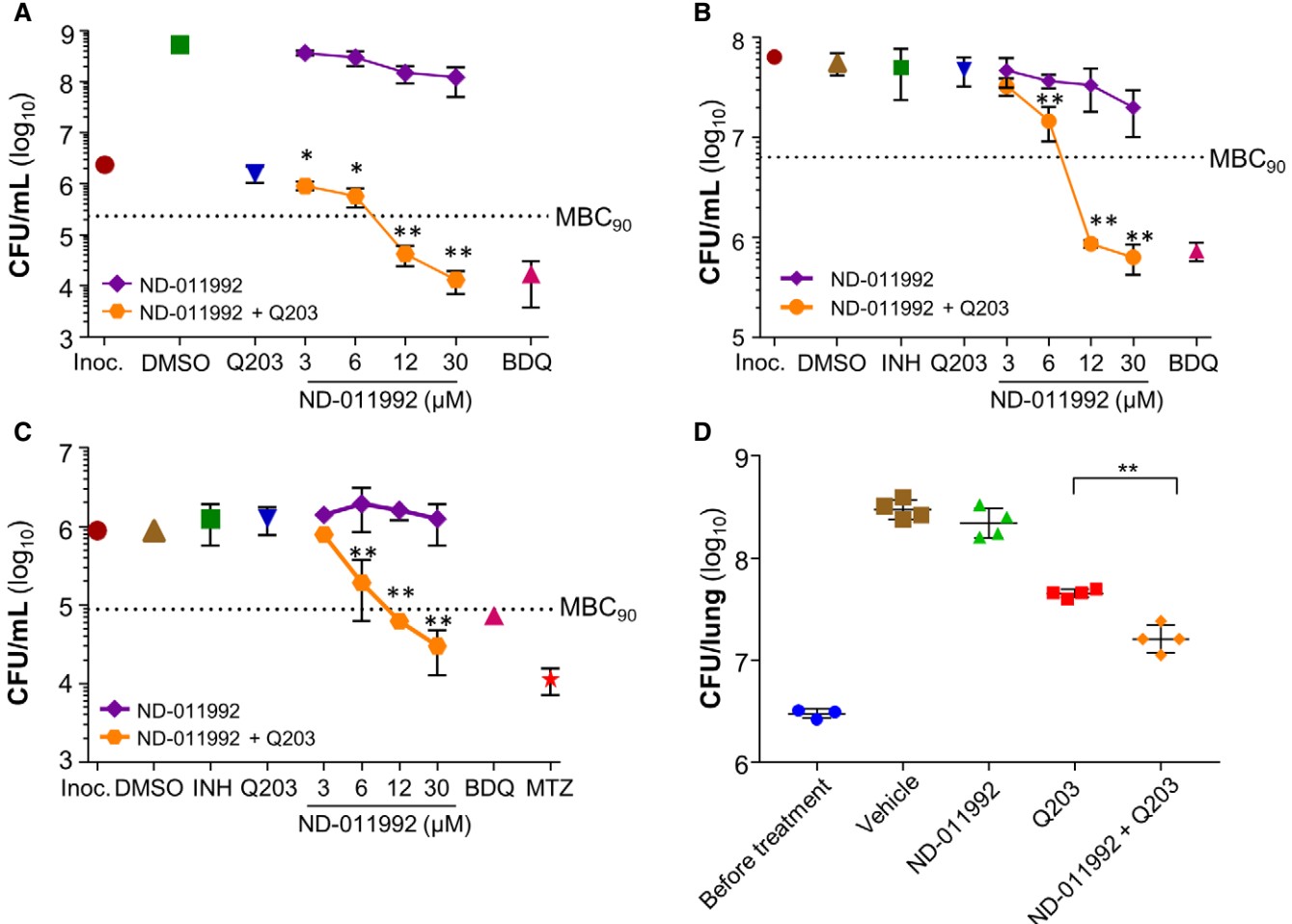

**Figure 4.  The combination ND-011992/Q203 is bactericidal against replicating and non-replicating *M. tuberculosis* H37Rv and shows potency *in vivo*.**

A   Bactericidal activity of ND-011992 and ND-011992-Q203 combination in replicating *M. tuberculosis* H37Rv. Bedaquiline (BDQ) was used at 500 nM, and Q203 at 100 nM. Inoc.: starting inoculum; DMSO: vehicle control. Bacteria viability was determined by enumerating colony-forming units (CFU) on nutrient-agar plates after 15 days of incubation. Dotted line represents 90% reduction in CFU/mL compared with the initial inoculum at time 0. *$P < 0.05$; **$P < 0.01$ unpaired Student's t-test, one-tailed; $n = 3$; comparing Q203 alone *vs* Q203 + ND-011992. Exact *P*-values (from left to right): 0.0346, 0.0143, 0.00016, 0.000052.

B   Bactericidal activity of ND-011992 and ND-011992-Q203 combination in nutrient-starved *M. tuberculosis* H37Rv. Bedaquiline (BDQ) was used at 500 nM, isoniazid was used at 1 μM, Q203 was used at 100 nM, and ND-011992 was used at 3, 6, 12, and 30 μM. Bacteria viability was determined by enumerating colony-forming units (CFU) after 15 days of incubation. Dotted line represents 90% reduction in CFU/mL compared with the initial inoculum at time 0. **$P < 0.01$ unpaired Student's t-test, one-tailed; $n = 3$; comparing Q203 alone *vs* Q203 + ND-011992. Exact *P*-values (from left to right): 0.0098, 0.000027, 0.000067.

C   Bactericidal activity of ND-011992 and ND-011992-Q203 combination in *M. tuberculosis* H37Rv under hypoxia. BDQ was used at 500 nM, isoniazid at 1 μM, metronidazole (MTZ) at 200 μM, Q203 at 100 nM, and ND-011992 at 3, 6, 12, and 30 μM. Bacteria viability was determined by enumerating colony-forming units (CFU) after 15 days of incubation. Dotted line represents 90% reduction in CFU/mL compared with the initial inoculum at time 0.**$P < 0.01$ unpaired Student's t-test, one-tailed; $n = 3$; comparing Q203 alone *vs* Q203 + ND-011992. Exact *P*-values (from left to right): 0.0092, 0.00010, 0.00079.

D   Efficacy of ND-011992-Q203 combination treatment in a mouse model of acute tuberculosis. Bacterial load (CFU) was enumerated in the lungs of mice before treatment, and after 5 days of treatment with either the vehicle control (brown squares), Q203 at 5 mg/kg (red squares), ND-011992 at 25 mg/kg (green triangles), or ND-011992 at 25mg/kg with Q203 at 5 mg/kg (orange diamonds). **$P = 0.0006$ unpaired Student's t-test, two-tailed; $n = 4$; comparing Q203 alone *vs* Q203 + ND-011992.

Data information: Data are expressed as the mean ± SD of triplicates for each data point. Experiments in (A), (B), and (C) were performed with 3 technical replicates and independently repeated at least once. *In vivo* experiment in (D) was performed with 4 replicates for all treatment groups, and 3 replicates to elucidate bacterial load before initiation of treatment. The experiment was repeated once.

either the Cyt-*bcc:aa₃* or the Cyt-*bd* are expected to bring about a collapse of this synergistic lethality, implying that the FOR to the combination treatment could be higher compared with Q203 alone. However, using a classical approach supported by fluctuation analysis, we observed the addition of ND-011992 did not increase the FOR to Q203. Analysis of the drug susceptibility profile of eighteen spontaneous-resistant colonies revealed that the primary

mechanism of resistance to the drug combination is through mutations conferring resistance to Q203. The absence of mutations in the *CydABDC* operon or any other genes besides *QcrB* results in a very low FOR ($< 3.7 \times 10^{-10}$) to ND-011992. One reason that might account for this is the strict essentiality of the ND-011992-interacting residues for Cyt-*bd* function, or another underlying, off-target action of the compound that remains to be determined. Oxidative

phosphorylation is the main source of energy production for the viability and replication of *M. tuberculosis* (Cook *et al*, 2017; Cook *et al*, 2014). Even in conditions where oxygen or nutrients are scarce, the energy metabolism pathway remains essential in helping the pathogen adapt and survive (Rao *et al*, 2008; Berney & Cook, 2010; Gengenbacher *et al*, 2010). The bactericidal activity of the drug combination in both non-replicating models (nutrient starvation and hypoxia) is a pivotal finding. While it may be intuitive to postulate that terminal oxidases are less relevant in environments with low oxygen tension, our finding that chemical inhibition of terminal oxidases is bactericidal even in hypoxic conditions confirmed the importance of the two terminal oxidases in this phenotypically drug-tolerant population. The *in vivo* study demonstrated that the addition of ND-011992 enhanced the potency of Q203. However, the enhanced potency is limited by ND-011992's less-than-optimal pharmacokinetic properties. Lead optimization is clearly required to optimize the potency of ND-011992 and improve its overall pharmacokinetics properties. Nevertheless, this result serves as a proof of concept that the synergism between a Cyt-*bd* inhibitor and Q203 is translational as far as in a murine model of infection and provides a case for Cyt-*bd* as an attractive target to be exploited. Future studies are needed to identify drugs capable of enhancing further the bactericidal potency of the combination Q203/ND-011992. Since our results indicate that the combination Q203/ND-011992 is at least as bactericidal as bedaquiline against replicating and non-replicating *M. tuberculosis*, it will be particularly interesting to test the bactericidal potency of the 3-drug combination first *in vitro*, and in animal models after the development of an improved version of ND-011992. Together, the results reported in this study demonstrate that a small-molecule inhibitor of the Cyt-*bd* acts synergistically with Cyt-*bcc:aa*$_3$ inhibitors such as Q203 to kill all *M. tuberculosis* subpopulations, including antibiotic-tolerant hypoxic ones, opening possibilities to shorten tuberculosis treatment time.

# Materials and Methods

### Strains and growth conditions

All mycobacteria strains were cultured in Middlebrook 7H9 medium (Becton Dickson and Company Limited, USA) supplemented with 0.05% Tween 80, 0.5% glycerol, and ADS enrichment (bovine serum albumin, D-glucose, and NaCl), or cultivated on Middlebrook 7H10 agar (Becton Dickson and Company Limited, USA) supplemented with OADC. *Mycobacterium tuberculosis* mc$^2$6230 was grown aerobically at 37°C in 7H9 media enriched with 10% OADC, 0.2% casamino acids, 24 mg/ml pantothenate, and 0.01% tyloxapol. Hypoxic cultures were prepared by harvesting bacteria at exponential phase and then washed in assay medium. The cultures were placed in a sealed hypoxic container for 4 days at 37°C. To prepare nutrient-starved cultures, bacteria were harvested in exponential phase and washed twice with DPBS (Dulbecco's phosphate-buffered saline) supplemented with 0.025% Tween 80. The culture was then adjusted to a density of OD$_{600}$ 0.15 and acclimatized for 2 weeks at 37°C.

The *M. tuberculosis* clinical isolates used in this study were a gift from Sebastien Gagneux. The *M. tuberculosis* cydAB knockout and corresponding complement strains used in this study were constructed in a previous study (Kalia *et al*, 2017). A *M. bovis* BCG strain overexpressing Cyt-*bd* was constructed by electroporating plasmid pMV262-*cydABDC* into the parental *M. bovis* BCG strain. The plasmid pMV262-*cydABDC* was constructed by incorporating the *cydABDC* operon, including 330bp upstream of the coding region, into the pMV262 vector. The *M. tuberculosis* ΔctaE-qcrCAB and corresponding complement strain were constructed in a previous study (Beites *et al*, 2019). The *M. smegmatis* ΔqcrCAB strain used in this study was constructed in a previous study (Chong *et al*, 2020). Prior to the start of all experiments, replicating cultures were harvested at logarithmic phase, washed to remove glycerol from its media, and diluted to specified cell density according to different experiments. Bedaquiline was purchased from Cellagen Technology LLC, USA. Isoniazid, carbonyl cyanide m-chlorophenyl hydrazine (CCCP), and metronidazole were obtained from Sigma-Aldrich Corp, USA.

### Chemistry

All anhydrous solvents, reagent grade solvents for chromatography, and starting materials were purchased from either Aldrich Chemical Co. (Milwaukee, WI) or Fisher Scientific (Suwanee, GA) unless otherwise noted. General methods of purification of compounds involved the use of 40–63 μm, 60 Å silica gel (SiliCycle, https://www.silicycle.com), and/or recrystallization. The reactions were monitored by TLC on precoated Merck 60 F254 silica gel plates and visualized using UV light (254 nm). All compounds were analyzed for purity by HPLC and characterized by $^1$H and $^{13}$C NMR using a Bruker DPX Avance I NMR Spectrometer (300MHz) and/or a Bruker Ascend Avance III HD Spectrometer (500 MHz). Chemical shifts are reported in ppm (δ) relative to the residual solvent peak in the corresponding spectra (chloroform δ 7.27), and coupling constants (J) are reported in hertz (Hz) (where, s = singlet, d = doublet, dd = double doublet, m = multiplet) and analyzed using ACD NMR data processing. $^{19}$F NMR were run without a standard and are uncorrected. Mass spectra values are reported as m/z. The melting point was measured on a Buchi B-545 and is uncorrected, and benzoic acid was used as a standard (Mp = 121.8–122.3°C).

Liquid chromatography-mass spectrometry method was performed on an Agilent 1290 infinity coupled to Agilent 6538 Ultra High Definition Quadrupole Time of Flight (UHD-QToF) instrument. A separation was achieved by using reverse-phase Waters Acquity UPLC HSS T3 1.8 μm (2.1 × 100 mm) column from Waters (Milford, USA). All solvents were purchased from Fischer Scientific LC-MS Optima grade solvents. Water containing 0.1% formic acid was used as mobile phase A, and acetonitrile containing 0.1% formic acid was used as mobile phase B. The injection volume was set at 1 μl. Samples were injected in a gradient of 95% mobile phase A and 5% mobile phase B in the initial condition to 5% mobile phase A and 95% mobile phase B in 10 min. The eluent was held at that composition for an additional 3 min and switched back to the initial condition at 12 min. The MS data acquisition was performed from 50 to 1000 m/z at 1.0 spectra/s scan rate. The source gas temperature was set at 350°C with a flow of 8 l/min. The nebulizer gas was set at 55 psig. The capillary voltage was set at 3,500 volts with fragmentor at 100, skimmer at 45 and octopole RF 500 volts. Prior to sample runs, the instrument was calibrated using Agilent low-mass calibrant solution.

The data collected in Agilent LC-MS were analyzed using Agilent Mass Hunter software for HRMS calculation.

## Synthesis of N-(4-(4-(trifluoromethyl)phenoxy)phenyl)quinazolin-4-amine, ND-011992

ND-011992

4-Chloroquinazoline (CAS: 5190-68-1, 700 mg, 4.3 mmol), 4-(4-(trifluoromethyl)phenoxy)aniline (CAS: 57478-19-0, 1.08 g, 4.3 mmol) and potassium carbonate (587 mg, 4.3 mmol) were dissolved in 15 ml of DMSO. The reaction was heated to 110°C for 12 h. The reaction mixture was concentrated to dryness, and the residue was dissolved in $CH_2Cl_2$ and washed with 5% acetic acid solution (2×), water, and brine. The organic phase was collected, dried over sodium sulfate ($Na_2SO_4$), filtered, and then concentrated *in vacuo*. Crude material obtained was purified by silica gel column chromatography with a 50% ethyl acetate: $CH_2Cl_2$ solvent system to give 1.27 g (78%) of *N*-(4-(4-(trifluoromethyl)phenoxy)phenyl)quinazolin-4-amine. $^1H$ NMR (300 MHz, $CDCl_3$) δ ppm 8.75 (s, 1H), 7.93 (dd, J = 8.3, 1.3 Hz, 2H), 7.87–7.74 (m, 2H), 7.74 (s, 1H), 7.58 (dd, J = 8.5, 7.1 Hz, 4H), 7.16–7.02 (m, 4H). $^{13}C$ NMR (125 MHz, $CDCl_3$) δ ppm 160.6, 157.5, 154.9, 152.3, 150.1, 134.6, 133.1, 129.1, 127.2 (q, J = 2.7 Hz), 126.8, 124.9 (q, J = 27.3 Hz), 123.7, 122.3 (q, J = 257.0 Hz), 120.6, 120.1, 117.7, 115.0. $^{19}F$ NMR (282 MHz, $CDCl_3$) δ ppm −61.72 (s, 3F). HRMS (EI), M + 1 calcd. for $C_{21}H_{15}F_3N_3O$, 382.1167; found 382.1157, HPLC $t_{R=}$ 5.928 min, Purity = 100%, Mp = 184.2–184.5°C.

## Quantitation of intracellular ATP level

The BacTiter-Glo™ Microbial Cell Viability Assay (Promega, USA) was used for quantitation of ATP content of bacteria cultures. Bacteria density was adjusted to $OD_{600}$ 0.05. 100 µl of culture was aliquoted into each well on a 96-well white plate, and drug compounds were subsequently added into each well to make up to corresponding concentrations. DMSO concentration was kept at 1% across all wells. The plates were incubated at 37°C for 15 h. Subsequently, BacTiter-Glo reagent was added to each well and incubated further for 12 min. The luminescence of each plate was measured using BioTek Cytation 3 Cell Imaging Multiple-mode reader. Data analysis was performed on GraphPad Prism 7 software.

## ATP synthesis assay in inverted-membrane vesicles (IMVs)

*M. tuberculosis* mc$^2$6230 were harvested at an $OD_{600}$ of 0.8 and IMVs made using the method described by Yano et al with slight modifications (Yano et al, 2011). Briefly, bacteria were lysed by 5 cycles of bead beating (1 min bead beating at 15,800 rpm followed by 2-min cooling on ice) in a Biospec Bead Beater (Bartlesville, USA). The samples were then centrifuged at 59,860 *g* for 210 min. After snap-freezing in liquid nitrogen, aliquots were stored at −80°C until use. Bradford reagent was used to estimate protein content which was shown to be 5 mg/ml.

All IMV experiments were carried out at 37°C in the same buffer that the IMVs were prepared in (10 mM HEPES, 50 mM KCl, 5 mM $MgCl_2$, 10% glycerol, pH 7) unless otherwise specified. ATP synthesis was measured using luciferase/luciferin (Roche Bioluminescence Assay Kit CL II). According to the kit protocols, luciferase/luciferin mix was combined with equal portions of buffer, IMVs (final concentration of 35 µg/ml), 250 mM NADH, 50 mM ADP, and 5 mM phosphate. Luminescence was monitored on a BioTek Synergy H4 plate reader in a 384-well plate. For the drug-treated wells, the IMVs were pre-incubated with drugs for 10 min. Negative control lacked NADH (electron donor). The rate of ATP synthesis was determined by calculating the rate of change in luminescence over 15–30 min following the addition of all reagents.

## Methylene blue assay

Stock methylene blue solution (0.5%) was purchased from Sigma-Aldrich. Two-milliliter screw-cap glass vials were filled with 1.8 ml of mycobacterial culture ($OD_{600}$ 0.3) in the presence of the test drugs. The cultures were incubated in the presence of the drugs for 6 h at 37°C before the addition of methylene blue dye at a final concentration of 0.001%. Upon addition of methylene blue, all vials were tightly closed and incubated at 37°C in a hypoxic jar. The vials were removed from the incubator after 96 h for image recording.

## Oxygen consumption assays using the Seahorse XFe96 analyzer

Agilent Seahorse cell culture microtiter plates were coated with 22.4 µg/ml Cell Tak (Corning®). *M. bovis* BCG was harvested at exponential phase and washed and adjusted to a concentration of $OD_{600}$ 0.4 in unbuffered 7H9. 50 µl of culture was added to each well and centrifuged to adhere the bacteria to the microtiter plate. The basal oxygen consumption rates (OCR) of *M. bovis* BCG were measured using the Seahorse XFe96 Analyzer (Agilent). The oxygen consumption rate (OCR) was derived from 4 min of continuous measurements. Each data point was interspaced with 3 min of continuous mixing. Compounds tested were injected real-time through the injection port. All data are analyzed using the Wave Desktop 2.6.0.31 as well as the GraphPad Prism 7 software.

## Oxygen consumption assays using the MitoXpress® probe

MitoXpress® Xtra was purchased from Luxcel Biosciences Ltd. Prior to the experiment, the probe was reconstituted in 1 ml sterile water. Cultures were diluted to $OD_{600}$ 0.5, and 150 µl was aliquoted into each well on a 96-well black-wall, clear-bottom plate. 1.5 µl of test compounds was added to corresponding wells, and 10 µL probe was added to each well. The wells were sealed off with two drops of high sensitivity oil and incubated at room temperature for 10 min before placing the plate in Cytation-3 Cell Imaging Multiple-mode reader (BioTek, USA). Dual-read time-resolved fluorescence (excitation: 380 nm, emission: 650 nm) with two integration windows (30-µs delay-30-µs measurement time; 30-µs delay-70-µs measurement time) was measured every 5 min for 6 to 8 h at 37°C. The data from dual-read time-resolved fluorescence were used to calculate the

fluorescence lifetime using the following transformation according to the manufacturer's manual. Data analysis was performed on GraphPad Prism 7 software.

## Oxygen consumption assay in Inverted-Membrane Vesicles (IMVs)

IMVs were prepared as previously described (Hards *et al*, 2018; Hards *et al*, 2019). 5 g (wet weight) of *M. smegmatis* cells, grown as previously described (Hards *et al*, 2015), was resuspended in buffer (50 mM Tris (pH 7.5), 5 mM $MgCl_2$ with 20 mg/l DNAse I and 1 Roche Complete Mini Protease inhibitor tablet per 50 ml). Cells were resuspended to approximately 0.125 [g wet weight cells]/ml. Cells were disrupted by two passages through a Stansted Fluid Power French pressure cell at 20,000 psi. Cell debris was removed by discarding the pellet after centrifugation (10 min, 15,000 *g*). The supernatant was centrifuged (45 min, 200,000 *g*) to collect IMVs. The membrane pellets were resuspended to a protein concentration of 25 mg/ml (as determined by a BCA assay (Sigma) with BSA as a protein standard) in buffer + 10% glycerol and stored in aliquots at −80°C until required.

Oxygen consumption was measured as previously described (Heikal *et al*, 2016; Pecsi *et al*, 2014), with the exception that an Oroboros O2k fluorometer was used with the previously described modifications (Saw *et al*, 2019). Oxygen consumption of IMVs was stimulated with 1 mM NADH. The Oroboros O2k Titration Injection Micropump module (TIP2k-Module) was used to titrate a 1 mM stock of ND-011992 from 1 nl/ml to 25 μl/ml in a stepwise manner. At least 30 s was allowed between each step to stabilize the oxygen consumption rate. An optimal IMV concentration (typically 0.125 mg/ml) was determined prior to each experiment, so that the oxygen in the cell was not fully depleted during the titration. Each titration was performed side-by-side with a titration of the DMSO vehicle, to exclude the effects of the vehicle on oxygen consumption rates. Other inhibitors were manually injected with Hamilton syringes, prior to the titration, as appropriate.

## Determination of minimum inhibitory concentration (MIC)

$MIC_{50}$ were determined as previously described (Kalia *et al*, 2017). 200 μl of mycobacteria culture ($OD_{600}$ 0.005) was dispensed into each well on 96-well flat bottom plates. 2 μl of drugs is dispensed into each corresponding well. The assay plates were then incubated for 5–6 days at 37°C. The effect of various treatment on bacteria growth was determined by measuring turbidity at 600 nm on a BioTek Cytation 3 Cell Imaging Multiple-mode reader.

## RNA Isolation

H37Rv and H37RvΔ*cydAB* were grown to an $OD_{600}$ of 0.2. Drugs were added, as indicated, at 50 nM Q203, 5 μM ND-011992. Samples of bacterial cultures were harvested in triplicate at appropriate times. 10 ml of culture was pelleted and the supernatant decanted. The cell pellet was resuspended in 1 ml of TRIzol (Invitrogen) and incubated overnight at 4°C, before storage at −80°C. The suspension was transferred to Fast-Prep Blue Cap tubes and processed twice for 45 s at speed 6 in a Fast-Prep apparatus (MP Biomedicals). After a brief incubation on ice, the sample was centrifuged at (12,000 *g* for

10 min), and the supernatant (< 750 μl) was processed for purification (described below).

## RNA sequencing

RNA from the clarified cell lysate samples was purified by TRIzol-chloroform precipitation. Briefly, 200 μl of chloroform was added to 700 μl of cell lysate, mixed well by shaking, and allowed to incubate at RT for 5 min. Samples were then centrifuged at 16,200 *g* for 15 min at 4°C, and 400 μl of the top aqueous layer was carefully collected. Then, 40 μl of 3 M sodium acetate pH 5.2 and 400 μl of isopropanol were added, mixed, incubated at RT for 15 min, and centrifuged as before. The supernatant was pipetted off and discarded, and the pellet was washed three times with ice-cold 70% EtOH, and centrifuged at 16,200 *g* for 5 min. After removing all remaining EtOH, the samples were air-dried for 10 min. Finally, the RNA pellet was resuspended in 30 μl of nuclease-free water and the concentration was determined by NanoDrop (Thermo Fisher). After purification, remaining DNA was removed from the samples using the DNA-Free Turbo DNase Kit (Ambion) and RNA integrity was check on a Bioanalyzer RNA 6000 Pico chip (Agilent). The ribosomal RNA (rRNA) was removed using the Bacterial RiboZero kit (Illumina), and removal was confirmed by Bioanalyzer as above. RNA-seq library preparation was conducted with the TrueSeq Stranded mRNA Library Prep (Illumina) according to manufacturer's instructions, and the library was sequenced on a NextSeq 500 (Illumina) using the NextSeq 500/550 HO V2 (75 cycles) (Illumina). The output was then analyzed as previously described (Benjak *et al*, 2015).

## UV spectral analysis of inverted-membrane vesicles (IMVs)

Wild-type (WT) *M. smegmatis* $mc^2155$- and the corresponding Δ*qcrCAB* mutant strain were grown to mid-log phase in LB media supplemented with Tween 80 (0.05%) by shaking at 180 rpm, 37°C. The cells were lysed on ice by sonication with an ultrasonic homogenizer (Bandelin, Berlin, Germany, KE76 tip) for 3 × 1 min in buffer A (50 mM MOPS/NaOH, pH 7.5, 2 mM $MgCl_2$, 2 mM Pefabloc$^{SC}$, and 50 μl/100 ml of DNase). After sonication, bacteria were broken by three passages through ice-cooled microfluidizer LM20 (Analytik, Cambridge, UK) at 15,000 psi. The cell lysate was then centrifuged at 15,000 *g* for 20 min at 4 °C (Eppendorf, Hamburg, Germany) to remove cell debris and non-lysed cells. The resulting supernatant was subjected to an ultracentrifugation step at 27,000 *g* for 40 min at 4°C. The supernatant was collected and centrifuged at 160,000 *g* for 60 min at 4°C as described by Kamariah *et al* (2019). The pellet, containing the IMVs, was resolved in buffer A containing 15% glycerol, liquid nitrogen frozen, and stored at −80°C. The protein concentration of the vesicles was determined by the bicinchoninic acid assay (BCA; Pierce, Rockford, IL, USA).

Parental *M. smegmatis* $mc^2155$- and the Δ*qcrCAB* mutant IMVs were analyzed with an Amersham Biosciences Ultrospec 2100 pro UV-visible absorption spectroscopy (New Jersey, USA) by recording spectra from 420 nm to 650 nm. To generate a defined oxidized state, 100 μM potassium ferricyanide was added to the IMVs adjusted at a protein concentration of 2 mg/ml. After collecting the spectra of the oxidized IMVs, 2 mM of NADH was added to reduce complex I and the respective cytochrome oxidase complexes within the WT and the *M. smegmatis* Δ*bcc* mutant IMVs. The UV-spectra were recorded

immediately after addition of NADH (0 min) and after 3 min, confirming that the IMVs are functional in electron transfer. The difference spectra of the reduced minus the oxidized forms were then computed. To study the effect of drugs, the respective compounds were added after the addition of the oxidant and before the addition of the electron donor NADH.

**Determination of frequency of resistance**

*M. tuberculosis* H37Rv or *M. bovis* BCG culture was harvested at exponential phase and concentrated by centrifugation to a cell density of approximately $5 \times 10^9$ CFU/ml. 7H10 agar was supplemented with drugs according to testing conditions. Each agar plate was inoculated with 200 μL of culture and subsequently incubated at 37°C for at least 21 days.

**Fluctuation analysis**

The assay was conducted as previously described (Ford *et al*, 2013). *M. tuberculosis* cultures were grown to exponential phase and then diluted to approximately $10^3$ CFU/ml. 4 ml of said culture was dispensed into $24 \times 30$ ml inkwell bottles, and incubated at 37°C for 11–14 days, until an $OD_{600}$ of 0.8–1.0 was reached. 4 bottles are diluted for total CFU enumeration. The other 20 cultures were then centrifuged and concentrated such that each culture was entirely plated onto a drug-supplemented 7H10 agar plate. The plates were incubated for 21–30 days.

**Drug susceptibility test on solid medium**

7H10 agar supplemented with various concentrations of drugs was prepared on quadrant plates. *M. tuberculosis* cultures were grown to exponential phase and diluted to approximately $10^6$ CFU/ml. 50 μl of culture was plated on each quadrant. The plates were incubated for at least 30 days. The minimum inhibitory concentration (MIC) is taken as the lowest concentration at which there was no visible growth.

**Bacterial viability assay**

Cultures were adjusted to an $OD_{600}$ of 0.005 and aliquoted into 24-well plates. Test compounds were dispensed into each well. The plates were incubated at 37°C. Bacteria viability was determined by enumerating colony-forming units (CFU) after plating on agar plates. The number of colonies was counted after agar plates were incubated at 37°C for 14–20 days.

**Mouse experiments**

Protocols for mouse studies were approved by the Institutional Animal Care and Use Committee of the National University of Singapore (Protocol # R15-01122). Female C57BL/6 mice (InVivos, Singapore) were infected via the intranasal route infection at a dose of $1.5 \times 10^5$ CFU per animal. Drug dosing was initiated 5 days post-infection. Drugs were formulated in 20% D-α-tocopherol polyethylene glycol 1,000 succinate (TPGS) + 1% DMSO and administered via gavage for 5 consecutive days. Q203 was used at 5 mg/kg, and ND-011992 was used at 25 mg/kg. Mice were sacrificed 5 days after

**The paper explained**

**Problem**
Tuberculosis is the leading cause of death from a single infectious agent. The emergence of multi- and extensively drug-resistant (M/XDR) tuberculosis complicates the global effort to control the spread of the disease. Due to the lack of effective drugs, infections caused by M/XDR tuberculosis have poor prognoses, advocating for the development of novel therapeutics. Telacebec (Q203), a drug candidate with demonstrated efficacy in human clinical trial phase 2a, inhibits bacterial growth by inhibiting the cytochrome $bcc:aa_3$, the major mycobacterial terminal oxidase. However, the bactericidal potency of Q203 is limited by the presence of an alternate terminal oxidase, the cytochrome *bd* oxidase. From the observation that the two terminal oxidases are jointly required for the survival of both replicating and antibiotic-tolerant *Mycobacterium tuberculosis*, we developed a facile screening strategy to identify a small molecule inhibitor of the Cytochrome *bd* oxidase, with the aim of boosting the effectiveness of Q203 using combination therapy.

**Results**
Through the screen of a focused library of small molecules, we identified ND-011992 as a putative cytochrome *bd* oxidase inhibitor. We demonstrated that ND-011992 and Q203 synergistically inhibited oxygen consumption and ATP synthesis in the pathogen. The drug combination was active against clinical isolates of various phylogenetic lineages, against MDR and XDR isolates, and killed replicating and antibiotic-tolerant non-replicating mycobacteria *in vitro*. Furthermore, the drug combination achieved better killing than Q203 alone in a mouse model.

**Impact**
The finding that dual inhibition of the two terminal oxidases kills *M. tuberculosis* from various phylogenetic origins and irrespective of their drug resistance profile opens the possibility of a new treatment strategy against MDR- and XDR-TB. The low frequency of resistance to ND-011992-Q203 and the bactericidal efficacy against antibiotic-tolerant, non-replicating mycobacteria suggest that inhibitors of the terminal oxidases could be part of a short sterilizing drug combination for tuberculosis.

the end of drug treatment to determine the bacterial loads in the lungs and spleens by CFU enumeration.

**Statistical analysis**

For animal studies, sample size calculations were not performed; 3 to 4 mice were randomly allocated to each experiment group, and investigators were not blinded. The experiment was successfully reproduced to demonstrate the consistency of findings. For all experiments, samples were considered to be normally distributed, and the type of statistical tests was appropriately chosen. Error bars shown in graphs indicate the standard deviation within the group. To ensure the groups that are being statistically compared have similar variance, *F*-test was conducted.

# Data availability

The RNA-seq data from this publication have been deposited to the Gene Expression Omnibus (GEO) database (https://www.ncbi.nlm.

nih.gov/geo/query.acc.cgi?acc=GSE159080) and assigned the identifier GSE159080.

**Expanded View** for this article is available online.

## Acknowledgments

This work was supported by the Singapore Ministry of Health's National Medical Research Council under its Cooperative Basic Research Grant (Project Award NMRC/CBRG/0083/2015, to K.P.), the National Research Foundation (NRF) Singapore under its NRF Competitive Research Programme (Project Award Number NRF–CRP18–2017–01, to G.G. and K.P.), by a Nanyang President's Graduate Scholarship (to B.S.L.), by the National Institutes of Health USA (R01 AI139465 to M.B., F30 AI138483 to E.J.H., R37 AI054193 to M.J.M., and R01 AI137043 to A.J.C.S.), by Potts Memorial foundation (awarded to M.B.), and by an IGS Premium Scholarship, Institute of Technology in Health and Medicine at NTU (to S.M.S.C.). K.H. and G.M.C. were funded by Marsden grants awarded by the Royal Society of New Zealand and a Health Research Council of New Zealand grant. A.J.C.S. is funded by the South African Medical Research Council. The graphical abstract was created with BioRender.com.

## Author contributions

KP supervised the project; BSL, NPK, MJM, and GCM designed or performed the whole-cell screen; BSL and NPK designed and performed growth inhibition assays, and *in vitro* bactericidal experiments; BSL, NPK, KH, and GMC designed and performed the oxygen consumption assays; EJH, MB, JC, XJ, CAE, DS, BSL, KP, KH, GMC, ES, SMSC, MSSM, and GG designed or performed target validation assays; JSM and AJCS performed testing against MDR- and XDR-TB clinical isolates and performed target validation experiments; BSL designed and performed the spontaneous resistance mutant cultivation and fluctuation analysis experiments; VHK and SA designed and performed the *in vivo* experiments; GCM and MJM provided screening library and synthesized compounds, including ND-011992; BSL and KP wrote the manuscript with contribution from other authors. All authors analyzed the results.

## Conflict of interest

M. J. M., G. C. M., and K. P. are named as inventors on a patent application related to this work: patent #WO2020128981, "Discovery of bd oxidase inhibitors for the treatment of mycobacterial diseases". Hsiri Therapeutics has licensed this technology. M. J. M. and G. C. M. own equity in Hsiri Therapeutics. M. J. M. is an employee of Hsiri therapeutics. G. C. M. and K. P. are consultants to Hsiri therapeutics. Hsiri Therapeutics did not fund this study and was not involved in study design or interpretation.

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
