## [Review Process File · EMBO Molecular Medicine]

Dual inhibition of the terminal oxidases eradicates antibiotic-tolerant *Mycobacterium tuberculosis*

Bei Shi Lee, Kiel Hards, Curtis Engelhart, Erik Hasenoehrl, Nitin Kalia, Jared Mackenzie, Ekaterina Sviriaeva, Shi Min Sherilyn Chong, Malathy Sony Manimekalai, Vanessa Koh, John Chan, Jiayong Xu, Sylvie Alonso, Marvin Miller, Adrie Steyn, Gerhard GRÜBER, Dirk Schnappinger, Michael Berney, Gregory Cook, Garrett Moraski, and Kevin Pethe

DOI: 10.15252/emmm.202013207

Corresponding authors: Kevin Pethe (kevin.pethe@ntu.edu.sg)

Review Timeline:

Submission Date:	4th Aug 20
Editorial Decision:	28th Aug 20
Revision Received:	6th Oct 20
Editorial Decision:	20th Oct 20
Revision Received:	28th Oct 20
Accepted:	30th Oct 20

Editor: Zeljko Durdevic

Transaction Report:

28th Aug 2020

Dear Dr. Pethe,

Thank you for the submission of your manuscript to EMBO Molecular Medicine. We have now received feedback from the three reviewers who agreed to evaluate your manuscript. As you will see from the reports below, all three referees are overall supportive of the study but also raise some concerns that should be addressed in a revision of the present manuscript. After cross-commenting exercise it became clear that there is no need for further experimentation. However, addressing all the the reviewers' concerns in writing will be necessary for further considering the manuscript in our journal. Furthermore, if you wish to add additional experiments to further strengthen your manuscript you are welcomed to do so. Please be aware that the acceptance of the manuscript might entail second round of review depending on the extent and elaborateness of the revision.

EMBO Molecular Medicine encourages a single round of revision only and therefore, acceptance or rejection of the manuscript will depend on the completeness of your responses included in the next, final version of the manuscript. For this reason, and to save you from any frustrations in the end, I would strongly advise against returning an incomplete revision.

We realize that the current situation is exceptional on the account of the COVID-19/SARS-CoV-2 pandemic. Therefore, please let us know if you need more than three months to revise the manuscript.

I look forward to receiving your revised manuscript.

***** Reviewer's comments *****

Referee #1 (Remarks for Author):

The authors describe in this paper that they identified a putative Cyt-bd inhibitor named ND-011992 through a facile whole-cell screen method. It synergistically inhibits ATP homeostasis and oxygen consumption in combination with Q203, an clinical inhibitor of Cyt-bcc:aa3, whilst ND-011992 is ineffective on its own. Further comparative analysis from transcriptomics, OCR sensitivity, bacterium growth and difference spectrum of hemes through gene knock-out and complementation provides multiple evidence that ND-011992 targets Cyt-bd. This is not an easy finding since identification of Cyt-bd inhibitors is not straightforward. Then, the authors showed that ND-011992 has a low spontaneous-resistance mutation frequency and is active against pan-susceptible clinical isolates and drug resistant clinical isolates. The most significant and exciting point in this paper is that ND-011992-Q203 combination is bactericidal against replicating mycobacteria and antibiotic-tolerant persisters. They finally demonstrated that ND-011992-Q203 combination enhanced the potency of Q203 in a mouse model, though lead optimization is required for ND-011992.

The authors were aware of the great concern of MDR and XDR TB cases and the lack of ways to rapidly eradicate infection. In this paper, they found a feasible way by targeting two terminal oxidases in oxidative phosphorylation to kill antibiotic-tolerant persisters. In my opinion, this work is valued by high medical impact and novelty. The paper is well written and well designed with adequate assays to support authors' conclusions. Before publishing this work, the authors should address the following points:

1. The authors proposed a way of dual inhibition of the terminal oxidases by the ND-011992-Q203 combination for more effective treatment of TB. Since there is already an approved drug bedaquiline which also targets the oxidative phosphorylation pathway. The author should have more analysis or discussion on comparison of the two drug-therapy ways. For example, what are the differences on the performance between ND-011992-Q203 and bedaquiline? Is there any potential advantage or complementary effect of ND-011992-Q203 combination on bedaquiline? Can we use ND-011992-Q203-bedaquiline three drug combination to enhance anti-TB efficacy?
2. The author should explain why treating with Q203 alone could enhance OCR compared to control in figure 1d and 1e, however in figure 1f, the OCR is reduced significant by Q203 compared to control? In figure S2b-e, there seems no difference between Q203 treatment and control.
3. In Page 8, the authors state that "escape mutants resistant to ND-011992 could not be isolated". This is lack of sufficient support. Though they found 18 escape mutants selected on ND-011992+Q203 are resistant to Q203, this cannot rule out the situation that they are resistant to both inhibitors. The authors should provide data like figure S5 by testing with ND-011992 alone.
4. Page 6 line 6 from bottom: how is 110-fold calculated? The data in figure 2c and Table S2 seem not match with this result.
5. Page 6 line 12-13 from bottom: these data are not consistent with Figure 2b.
6. Page 7 line 7 from top: a typo? 523 nm or 532 nm?
7. Page 7 line 9 from bottom: from {less than or equal to} 0.2 to 2 μ M
8. Page 12: the compound name ND-011992 should be consistent throughout the paper.
9. Page 14-15: use g rather than RPM.
10. Page 15: PMVs should be IMVs
11. Figure 1b,1d: provide the concentration of the inhibitors in the legend.
12. Figure 2a: the gene names are not readable. please enlarge this panel.
13. Figure 3: the order of panel should be adjusted to follow the citation order in the maintext. In the legend, line 7, Q203 is missing after 100 nm. You should keep one decimal space after the decimal point for all the data in panel d.
14. Figure S2c,2e: mark the significant of differences through statistic test.

15. Table S1: SD value is missing.

16. Table S2: is the value of 3.54 correct?

Referee #2 (Remarks for Author):

In their paper entitled "Dual inhibition of the terminal oxidases eradicates antibiotic-tolerant *Mycobacterium tuberculosis*", Lee et al. report on the use of a whole-cell screen approach to identify a small molecule inhibitor of cytochrome bd (ND-011992), which, when used in combination with telacebec (Q203), an inhibitor of the cytochrome bcc:aa3 terminal oxidase, shows promising activity against replicating mycobacteria and antibiotic-tolerant persisters in various in vitro stress models and in a mouse model of acute TB. The paper addresses a very important area of TB pathogenesis and the need for developing novel strategies to target Mtb persisters, which are believed to account for the difficulty in eradicating infection with the current standard-of-care regimen. Strengths of the study include the focus on oxidative phosphorylation, which appears to be a critical pathway for Mtb viability under both replicating and nonreplicating conditions, as well as the extensive studies undertaken to characterize the on-target effects of the compound, including studies with deletion and overexpression strains and transcriptomics analyses. Also, the activity of ND-011992 was tested against various clinical isolates, including drug-susceptible and drug-resistant isolates. Several issues outlined below diminish the overall enthusiasm for the paper in its current form.

Major comments:

1. The Introduction and conclusions focus on Mtb, but various experiments have been performed in three different species (*M. tuberculosis*, *M. bovis* BCG, or *M. smegmatis*) without a clear justification in the main text, or even a clear indication of which strain is being used (even in the figure legends it is sometimes unclear, e.g., Fig. 1e). It is not clear, for example, why the inverted membrane vesicle studies were performed in *M. smegmatis* (Supplementary Figure 4). Also, the authors do not address why there is a significant discrepancy in the IC₅₀ of ND-011992 required to deplete ATP levels in combination with Q203 in BCG vs. Mtb (Figure 1b and Supplementary Figure 1a), as well as the minimum FIC index in the two species (Supplementary Figures 1b and 1c). Does the compound show higher affinity for Cyt-bd from BCG vs. Mtb? What is the sequence homology and structural similarity between the two genes/proteins?
2. Based on experience with other chemical library screens, it is highly remarkable that ND-011992 was identified from a collection of 53 small molecules. How were these compounds selected? Were they in any way enriched for potential hits?
3. For the methylene blue assays, it would be helpful to have a media-only control to understand the background signal.
4. Fig 1 d-f: Indicate to what is %OCR normalized.
5. Fig 1f : This would be more convincing if the *cydAB* gene deleted and complemented were from the same species. Also, an H37Rv Δ *cydAB* strain is later used (Fig 3b), for which complementation data are lacking.
6. Fig 2a: Which genes are significantly regulated? This information could be included in the supplement. Also, it is stated that the transcriptional response of H37Rv treated with the Q203/ND-011992 combination was almost identical to the response of H37Rv Δ *cydAB* treated with Q203 alone, but it is not clear what % of genes are actually overlapping between these two groups. Also, what are the pathways represented by the ~200 genes that are differentially regulated in the mutant treated with Q203 alone vs. the WT treated with the two drugs?
7. Fig 4a-c: A Q203-alone control is not shown, so it is difficult to determine whether the activity of

the combination is truly synergistic, as described, or simply additive (ND-011992 has very little activity against actively replicating Mtb, as well as against nutrient-starved and hypoxic Mtb). Judging by the in vivo data (Fig 4d), the combination appears to be additive, but this can be formally calculated.

8. In vivo studies: The argument is made that pharmacological inhibition of both the cytochrome bcc:aa3 terminal oxidase and the cytochrome bd oxidase is a favorable strategy to target antibiotic-tolerant Mtb in order to shorten treatment length, however the animal model used is a very short-term model (5 days), in which the bacteria are actively replicating (and not antibiotic-tolerant). In addition, the study uses intranasal infection and C56BL/6 mice, which develop highly cellular lesions. A more appropriate model to truly evaluate the activity of this compound combination on antibiotic-tolerant Mtb would be a chronic infection model (initiation of treatment at least 4-6 weeks after (preferably aerosol) infection). Also, the C3HeB/FeJ mouse model would be preferable since, unlike C57BL/6 mice, it develops human-like necrotic granulomas, which, because of tissue hypoxia and other stress conditions, are believed to promote antibiotic tolerance in the extracellular bacteria found in the caseous core. Also, this model would offer a way of testing the penetration of these compounds into pathological lesions of clinical relevance.

Referee #3 (Remarks for Author):

Summary and General Comments

The manuscript "Dual inhibition of the terminal oxidases eradicates antibiotic-tolerant Mycobacterium tuberculosis" is an exciting study describing the identification of a cytochrome bd inhibitor which synergises with Q203 and they quite convincingly shows targets cytochrome bd oxidase and presents an excellent approach to targeting oxidative phosphorylation for tuberculosis treatment. The drug combination showed good activity in vitro and also in a murine model. I have some minor comments which are discussed below.

Minor points

- (1) In the abstract the authors state "The drug combination was bactericidal against replicating mycobacteria and antibiotic tolerant Persisters" however the experimental data shows that the drugs work on populations of cells in models which reduce the growth rate of Mtb. Whilst these experiments show the efficacy of the drugs against POPULATIONS of cells which are tolerant to other anti-TB drugs the experiments do not measure killing of the specialised sub-population of persister cells. To do this the authors would need to perform kill curves to test for changes in the biphasic kill curve (see more about this below). This just need rephrasing so that it doesn't mislead the reader. This also needs to be considered throughout the rest of the document.
- (2) It is interesting that the drug combination was more effective against BCG as compared with Mtb. Why do the authors think this is the case? It seems worthy of a line of discussion. And considering this its strange that for the identified resistant strain the authors only investigated BCG. Were the Mtb drug resistant strains also investigated?
- (3) For some of the experiments there needs to be some justification on the strain use as data seems to be presented from BCG, Mtb and M. smegmatis in Figure 1.
- (4) For the experiments where the authors measure the killing efficiency of the drug combinations in the Betts starvation model and a hypoxic model the authors pick one time point. Although there is an inoculum bar there doesn't seem to be a drug free control on figure 4b and 4c whereas there is a control on the growing cells (4a) and there is no Isoniazid control on figure 4a which are important and should be added. Im convinced that the drug combination works on these slow growing drug tolerant cells but without a time course the authors cannot comment on the effect on the persister population.

Referee #1 (Remarks for Author):

Before publishing this work, the authors should address the following points:

1. The authors proposed a way of dual inhibition of the terminal oxidases by the ND-011992-Q203 combination for more effective treatment of TB. Since there is already an approved drug bedaquiline which also targets the oxidative phosphorylation pathway. The author should have more analysis or discussion on comparison of the two drug-therapy ways. For example, what are the differences on the performance between ND-011992-Q203 and bedaquiline? Is there any potential advantage or complementary effect of ND-011992-Q203 combination on bedaquiline? Can we use ND-011992-Q203-bedaquiline three drug combination to enhance anti-TB efficacy?

We agree with this comment: it will be important to evaluate if bedaquiline (or other approved drugs) could further enhance the bactericidal potency of the Q203/ND-011992 combination. The following paragraph was added in the discussion section (Lines 340-345):

“Future studies are needed to identify drugs capable of enhancing further the bactericidal potency of the combination Q203/ND-011992. Since our results indicate that the combination Q203/ND-011992 is at least as bactericidal as bedaquiline against replicating and non-replicating M. tuberculosis, it will be particularly interesting to test the bactericidal potency of the 3-drug combination first in vitro, and in animal models after the development of an improved version of ND-011992.”

2. The author should explain why treating with Q203 alone could enhance OCR compared to control in figure 1d and 1e, however in figure 1f, the OCR is reduced significant by Q203 compared to control? In figure S2b-e, there seems no difference between Q203 treatment and control.

The experimental setting varied significantly between the experiments represented on those figure panels. In Figure 1D-E, the OCR measurements were performed on a Seahorse XFe96 Analyzer. The result reflects rapid changes immediately after the addition of Q203. The mechanisms that drive this initial spike in OCR detected with the Seahorse Xfe96 analyser is still unclear but has already been reported by some of us (Lamprecht *et al*, 2016). It likely reflects an immediate stress response upon chemical inhibition of the Cyt-*bcc-aa*₃ characterized by a transient increase in activity of the Cyt-*bd*. Interestingly, this phenomenon is also observed upon treatment with bedaquiline (Lamprecht *et al.*, 2016).

In the Figure EV2B-E (previously Supplementary figure 2b-e), the OCR was quantified using the MitoXpress® Probe based on readings collected over a few hours, and represent a cumulative, averaged OCR. The assay also included a period of 10 min preincubation before OCR is recorded. While the Seahorse XFe96 analyzer operates at constant oxygen concentration (the microchamber is mixed every few minutes to maintain high oxygen concentration (see Methods and Materials section), the MitoXpress assay does not. These differences in methodology may explain why an initial spike in OCR is not observed using the MitoXpress assay.

Figure 1F involved testing in inverted membrane vesicles (IMVs) from *Mycobacterium smegmatis*. The use of IMVs may enhance the potency of Q203 as the target QcrB is more readily accessible to chemical inhibition by Q203 and hence inhibition of OCR is the

dominant effect observed in these experiments. The following sentence was added to the main text (Lines 129-131):

“The use of IMVs may enhance the potency of Q203 as the target QcrB is more readily accessible to chemical inhibition by Q203 and hence amplifying the effect of OCR inhibition in these experiments.”

3. In Page 8, the authors state that "escape mutants resistant to ND-011992 could not be isolated". This is lack of sufficient support. Though they found 18 escape mutants selected on ND-011992+Q203 are resistant to Q203, this cannot rule out the situation that they are resistant to both inhibitors. The authors should provide data like figure S5 by testing with ND-011992 alone.

We initially assumed that the frequency of co-resistance was too low to think that the escape mutants resistant to Q203 could be resistant to ND-011992 as well. To address this concern, we conducted an additional experiment. It was previously reported that lansoprazole, a Cyt-*bcc:aa₃* inhibitor, remains potent against Q203-resistant mycobacteria (Rybniker *et al.*, 2015). After confirming that lansoprazole was potent against the 18 mutants selected on Q203 + ND-011992 (Table EV1), we tested the ability of ND-011992 to deplete intracellular ATP in the 18 escape mutant strains in the presence of a fixed dose of lansoprazole. Results confirmed that ND-011992 was still active in all isolated mutant strains, indicating that the escape mutants are susceptible to ND-011992 (Table EV1).

The following sentences were added to the main text (Lines 214-225):

*“All 18 M. bovis BCG colonies analyzed were highly resistant to Q203 ($MIC_{50} > 100$ nM) (Table EV1), suggesting that the primary mechanism of resistance to the drug combination is mediated by a mutation in QcrB (Pethe *et al.*, 2013). To exclude that any of the escape mutants were co-resistant to ND-011992, we made use of lansoprazole, an inhibitor of the mycobacterial cytochrome *bcc:aa₃* that retains efficacy against Q203-resistant spontaneous mutants (Rybniker *et al.*, 2015). Upon confirmation that lansoprazole retained efficacy against the spontaneous resistant mutants (Table EV1), we tested the inhibitory activity of a dose-range of ND-011992 in the same intracellular ATP assay used in the initial screen (Figure 1B), but in the presence of lansoprazole instead of Q203. ND-011992 depleted intracellular ATP levels at IC_{50} values comparable to those observed in the parental M. bovis BCG strain (Table EV1), indicating that the escape mutants remained fully susceptible to ND-011992.”*

4. Page 6 line 6 from bottom: how is 110-fold calculated? The data in figure 2c and Table S2 seem not match with this result.

The 110-fold increase was calculated using the data obtained in one of the experimental repeats. We amended to include a range of values calculated from experimental repeats (Lines 166-169):

*“Consistently, ND-011992 inhibited the growth of an H37Rv strain deficient in Cyt-*bcc:aa₃* expression (Beites *et al.*, 2019) at a concentration 33- to 110-fold lower compared to the concentration required to inhibit the growth of the parental strain (Figure 2C and Appendix Table S2).”*

5. Page 6 line 12-13 from bottom: these data are not consistent with Figure 2b.

The following correction has been made to reflect the correct values (Lines 160-162):

“Increasing the gene copy number of the cydABDC operon from a high-copy number plasmid caused an 18- to 31-fold shift in ND-011992 potency from 0.52 to 17.2 μM , as well as a 12-fold shift in OCR from 0.62 to 7.8 μM (Figure 2B).”

6. Page 7 line 7 from top: a typo? 523 nm or 532 nm?

The typographical error was corrected in the following sentence (Lines 177-179):

*“Similarly, NADH-driven reduction of the cytochrome bcc:aa₃ oxidase hemes (a at 443 and 600 nm, b at 432 and 563 nm, and c at 523 and 552 nm) was also clearly assigned for the parental *M. smegmatis mc² 155 IMVs* (Figure EV3A).”*

7. Page 7 line 9 from bottom: from {less than or equal to} 0.2 to 2 μM

The typographical error was corrected in the following sentence (Line 199):

“ND-011992 was potent at an MIC value ranging from \leq 0.2 to 1 μM against pan-susceptible clinical isolates...”

8. Page 12: the compound name ND-011992 should be consistent throughout the paper.

The subheading under Materials and Methods was corrected to the following:

“Synthesis of N-(4-(4-(trifluoromethyl)phenoxy)phenyl)quinazolin-4-amine, ND-011992”.

The schematic diagram under the same subsection was also updated to include the corrected compound name.

9. Page 14-15: use g rather than RPM.

The centrifuge speeds were changed to the unit of \times g in the following sentences:

Lines 527-528: *“Samples were then centrifuged at 16,200 \times g for 15 min at 4 $^{\circ}\text{C}$ and 400 μL of the top aqueous layer was carefully collected.”*

Lines 530-531: *“The supernatant was pipetted off and discarded and the pellet was washed three times with ice cold 70% EtOH, and centrifuged at 16,200 \times g for 5 min.”*

10. Page 15: PMVs should be IMVs

The following sentence was corrected (Lines 556-558):

“Parental *M. smegmatis* mc²155- and the Δ qcrCAB mutant *IMVs* were analyzed with an Amersham Biosciences Ultrospec 2100 pro UV-visible absorption spectroscopy (New Jersey, US) by recording spectra from 420 nm to 650 nm.”

11. Figure 1b,1d: provide the concentration of the inhibitors in the legend.

The legend of Figure 1 was corrected as follows:

“(B) Effect of ND-011992 on the intracellular ATP level in *M. bovis* BCG. The bacteria were treated with DMSO (blue triangle), 100 nM Q203 alone (green triangle), 20 μ M ND-011992 alone (brown square), and a dose-range of ND-011992 in the presence of Q203 (red squares) for 15 hours before quantification of intracellular ATP levels.” and “(D) Oxygen consumption assay in *M. bovis* BCG using the Seahorse XFe96 Extracellular flux analyser. 12 μ M ND-011992 was injected alone (green triangles), 100 nM Q203 (blue circles), or in combination with Q203 (red squares).”

12. Figure 2a: the gene names are not readable. please enlarge this panel.

Figure 2 was modified to improve readability of the gene names.

13. Figure 3: the order of panel should be adjusted to follow the citation order in the maintext. In the legend, line 7, Q203 is missing after 100 nm. You should keep one decimal space after the decimal point for all the data in panel d.

The panels in Figure 3 was reordered to reflect their citation order in the main text.

The figure legend was corrected as follows:

“Fluctuation analysis in *M. tuberculosis* H37Rv. *M. tuberculosis* Δ cydAB was plated on 7H10 containing 100 nM Q203 (red). Parental H37Rv was plated on 7H10 with 100 nM Q203 and 6 μ M ND-011992 or 2 μ g/mL rifampicin (blue). The corresponding values of the median frequency of resistance is indicated in the graph.”.

All MIC values in panel B (previously panel d) were standardised to the accuracy of one decimal place.

14. Figure S2c,2e: mark the significant of differences through statistic test.

Figures EV2C and EV2E (previously S2c and S2e) and their corresponding legends were updated to include the significance of differences.

15. Table S1: SD value is missing.

Appendix Table S1 was amended to include the standard deviation of the measurements.

16. Table S2: is the value of 3.54 correct?

Yes, the value is correct, complementation appeared to be partial in this set of experiment.

Referee #2 (Remarks for Author):

Several issues outlined below diminish the overall enthusiasm for the paper in its current form.

Major comments:

1. The Introduction and conclusions focus on Mtb, but various experiments have been performed in three different species (*M. tuberculosis*, *M. bovis* BCG, or *M. smegmatis*) without a clear justification in the main text, or even a clear indication of which strain is being used (even in the figure legends it is sometimes unclear, e.g., Fig. 1e). It is not clear, for example, why the inverted membrane vesicle studies were performed in *M. smegmatis* (Supplementary Figure 4). Also, the authors do not address why there is a significant discrepancy in the IC50 of ND-011992 required to deplete ATP levels in combination with Q203 in BCG vs. Mtb (Figure 1b and Supplementary Figure 1a), as well as the minimum FIC index in the two species (Supplementary Figures 1b and 1c). Does the compound show higher affinity for Cyt-bd from BCG vs. Mtb? What is the sequence homology and structural similarity between the two genes/proteins?

We agree that the rationale behind the use of different mycobacterial species should have been better explained throughout the manuscript.

For practical biosafety reasons, the assay was initially developed in *M. bovis* BCG. *M. bovis* BCG is an excellent surrogate mycobacterial species to study the terminal oxidases since its Cyt-*bcc:aa*₃ and Cyt-*bd* share 100% sequence similarity with the *M. tuberculosis* H37Rv counterparts (Brosch *et al*, 2007; Lew *et al*, 2011; Data ref: Garnier T., 2006; Data ref: Lew J. M., 2012). In addition, we have shown that *M. tuberculosis* and *M. bovis* BCG have a similar sensitivity to Q203, and deletion of *cydAB* led to comparable phenotypes (Kalia *et al*, 2017), supporting the use of *M. bovis* BCG as an acceptable surrogate. After obtaining key results in *M. bovis* BCG, some of the key findings were validated in *M. tuberculosis* H37Rv.

For added clarity, the species used for each experiment was mentioned in the text and in the figure legends. For consistency, the results presented in Figure 1C (obtained with *M. tuberculosis* H37Rv) was moved to Figure EV2A and substituted with the *M. bovis* BCG results (shown in Supplementary Figure 2a in the initial submission).

The reason why *Mycobacterium smegmatis* was used in the IMV experiments is because the model is well-established for assessing biochemical activities of the entire respiratory chain in mycobacteria, and to provide confirmation of target engagement in a membrane-only environment (Hards *et al*, 2015; Lu *et al*, 2015; Pecsí *et al*, 2014). Given that the *M. tuberculosis* Cyt-*bd* has high sequence identity with the *M. smegmatis* homologue (78% and 68% for CydA and CydB, respectively) we consider this biochemical system to be meaningful. The data obtained with *M. smegmatis* IMVs was in agreement with whole cell data for *M. tuberculosis* and *M. bovis* BCG.

To clarify the rationale behind the use of multiple mycobacteria species in the study, we have added the following sentences to the main text:

Lines 88-92: “For practical biosafety reasons, the assay was initially developed in *M. bovis* BCG. *M. bovis* BCG is an excellent surrogate mycobacteria to study the terminal oxidases since its Cyt-*bcc:aa*₃ and Cyt-*bd* share 100% sequence similarity with the *M. tuberculosis*

H37Rv counterparts (Brosch et al., 2007; Lew et al., 2011; Data ref: Garnier T., 2006; Data ref: Lew J. M., 2012)."

Lines 119-124: "*Biochemical assays were performed on purified inverted-membrane vesicles (IMVs) from Mycobacterium smegmatis mc²155 to obtain preliminary evidence that ND-011992 acts on the Cyt-bd in the lipid-rich environment of the IMVs. IMVs purified from M. smegmatis were used since protocols for their purification in high amount and subsequent characterization of the associated electron-transport chain are well-established (Hards et al., 2015; Heikal et al, 2016; Lu et al., 2015; Pecsí et al., 2014)."*

Figure legend of Figure 1E was corrected to the following to clarify the strain used in the experiment:

"(E) Dose-dependent inhibition of M. bovis BCG OCR by ND-011992 (in the presence of 100 nM Q203) measured on a Seahorse XFe96 analyser platform."

2. Based on experience with other chemical library screens, it is highly remarkable that ND-011992 was identified from a collection of 53 small molecules. How were these compounds selected? Were they in any way enriched for potential hits?

The compounds were selected from a long-standing NIH project that was carried out in the laboratory of Prof. Marvin Miller. Prof. Marvin Miller discovered several chemical series targeting QcrB (Moraski et al, 2012; Moraski et al, 2014; Tiwari et al, 2013). These 53 compounds were previously screened against *M. tuberculosis* H37Rv (MABA assay) and were selected based on their lack of potency, which is what we expected from a Cyt-bd inhibitor.

The following sentence was added (Lines 94-96).

"The small-molecules were selected from a larger library assembled during a NIH project focused on the design and discovery of novel small-molecules for tuberculosis treatment (Moraski et al., 2012; Moraski et al., 2014; Tiwari et al., 2013)."

3. For the methylene blue assays, it would be helpful to have a media-only control to understand the background signal.

As described in the manuscript, this assay is not quantitative but has the advantage to provide a rapid visual indication of the combined effect of Q203 + ND-011992 on oxygen consumption. As such, we have not added a medium-only control in the initial experiments. Instead, we relied on the Seahorse XFe96 analyser and the MitoXpress® probe to provide quantitative value of the extent of OCR inhibition by Q203 + ND-011992.

Nevertheless, we addressed this comment by performing a new experiment including a medium-only control in *M. bovis* BCG shown as Figure 1C in the revised manuscript. The same experiment could not be performed on *M. tuberculosis* because we do not have access to a BSL-3 laboratory since the beginning of the COVID-19 pandemic.

4. Fig 1 d-f: Indicate to what is %OCR normalized.

The figure legends of Figure 1D-F was edited to include the information:

“(D) Oxygen consumption assay in M. bovis BCG using the Seahorse XFe96 Extracellular flux analyser. 12 μ M ND-011992 was injected alone (green triangles), 100 nM Q203 (blue circles), or in combination with Q203 (red squares). For each condition, OCR readings were normalised to the last basal OCR reading before drug injection. OCR: Oxygen Consumption Rate (E) Dose-dependent inhibition of M. bovis BCG OCR by ND-011992 (in the presence of 100 nM Q203) measured on a Seahorse XFe96 analyser platform. For each condition, OCR readings were normalised to the last basal OCR reading before drug injection. (F) Effect of ND-011992 on the OCR of M. smegmatis IMVs using the Oroboros O₂k fluorespirometer. IMVs OCR from the parental strain (green triangles), Δ cydAB knockout (blue circles), and Δ cydAB complemented with M. tuberculosis CydABDC⁺ (red squares) energized with NADH were determined. Q203 was used at 1 μ M. 100% OCR was defined as the OCR of the untreated samples for each strain.”

5. Fig 1f: This would be more convincing if the cydAB gene deleted and complemented were from the same species. Also, an H37Rv Δ cydAB strain is later used (Fig 3b), for which complementation data are lacking.

We deemed the results convincing as they demonstrate that ND-011992 can indeed inhibit the M. tuberculosis Cyt-bd in the lipid-rich environment of the IMVs. This experiment was performed with all the appropriate controls providing the conclusive proof. The following sentence was added in the main text (Lines 124-127):

“To provide further evidence of target engagement of ND-011992 with the M. tuberculosis Cyt-bd in a lipid-rich environment, we prepared IMVs from M. smegmatis Δ cydAB mutant expressing the M. tuberculosis cydABDC operon.”

In Figure 3D (previously Figure 3b), we used the H37Rv Δ cydAB strain as a reference to compare the Fluctuation analysis of Q203 + ND-011992. Ideally, the H37Rv parental strain should have been used. However, it could not be done since, under the experimental conditions used to conduct the fluctuation analysis (at least 25 days of incubation at 37°C), a lawn would have been observed using a parental or Δ cydAB-complemented strain due to the bacteriostatic nature of Q203 and loss of potency on glycerol-supplemented agar plates (Kalia et al, 2019).

We added the following sentence (Lines 228-233):

“In this experiment, we used the H37Rv Δ cydAB strain to compare the frequency of resistance to Q203 + ND-011992 to a reference. The H37Rv parental strain could not be used due to the bacteriostatic nature of Q203 and the loss of potency of the drug on agar plates supplemented with glycerol (Kalia et al., 2019). In addition, rifampicin was added as an additional control to compare the frequency of resistance to Q203 + ND-011992 to an approved drug.”

6. Fig 2a: Which genes are significantly regulated? This information could be included in the supplement. Also, it is stated that the transcriptional response of H37Rv treated with the Q203/ND-011992 combination was almost identical to the response of H37Rv Δ cydAB treated with Q203 alone, but it is not clear what % of genes are actually overlapping between these two groups. Also, what are the pathways represented by the ~200 genes that are

differentially regulated in the mutant treated with Q203 alone vs. the WT treated with the two drugs?

In order to allow readers to determine which genes are differentially regulated in response to drug challenge, we have included a searchable Microsoft Excel document containing all of the processed differential gene expression data (Source data to Figure 2A).

To address questions about degree of overlap we have included the sentence (Lines 150-153):

“The majority of these genes were downregulated, and of the 891 downregulated genes in the combination-treated H37Rv strain, more than two-thirds (67.6%) were similarly downregulated in the Δ cydAB mutant challenged with Q203 alone.”

The extra 200 genes differentially regulated in the Q203-treated- Δ cydAB compared to combination-treated-H37Rv encompassed a wide diversity of pathways without any clear representation worthy of discussion here.

7. Fig 4a-c: A Q203-alone control is not shown, so it is difficult to determine whether the activity of the combination is truly synergistic, as described, or simply additive (ND-011992 has very little activity against actively replicating Mtb, as well as against nutrient-starved and hypoxic Mtb). Judging by the in vivo data (Fig 4d), the combination appears to be additive, but this can be formally calculated.

In Figure 4A-C, the Q203-alone control is shown in blue inverted triangles.

8. In vivo studies: The argument is made that pharmacological inhibition of both the cytochrome bcc:aa3 terminal oxidase and the cytochrome bd oxidase is a favorable strategy to target antibiotic-tolerant Mtb in order to shorten treatment length, however the animal model used is a very short-term model (5 days), in which the bacteria are actively replicating (and not antibiotic-tolerant). In addition, the study uses intranasal infection and C56BL/6 mice, which develop highly cellular lesions. A more appropriate model to truly evaluate the activity of this compound combination on antibiotic-tolerant Mtb would be a chronic infection model (initiation of treatment at least 4-6 weeks after (preferably aerosol) infection). Also, the C3HeB/FeJ mouse model would be preferable since, unlike C57BL/6 mice, it develops human-like necrotic granulomas, which, because of tissue hypoxia and other stress conditions, are believed to promote antibiotic tolerance in the extracellular bacteria found in the caseous core. Also, this model would offer a way of testing the penetration of these compounds into pathological lesions of clinical relevance.

We agree with the reviewer that additional models are required to test the potency of the combination Q203 + a cytochrome-*bd* inhibitor against persisters in an animal model.

We did not do it because, as explained in the manuscript, the pharmacokinetic properties of ND-011992 are sub-optimum. Many parameters such as potency, penetration in eukaryotic cells, volume of distribution, etc. need to be optimized to obtain a drug candidate compatible for testing in more complex *in vivo* models.

In the context of this article, we were pleased to observe some degree of potency of ND-011992 (in combination with Q203) in an acute mouse model of tuberculosis. As written in the text, chemical optimization is required to develop an advanced drug candidate.

Referee #3 (Remarks for Author):

I have some minor comments which are discussed below.

Minor points

(1) In the abstract the authors state "The drug combination was bactericidal against replicating mycobacteria and antibiotic tolerant Persisters" however the experimental data shows that the drugs work on populations of cells in models which reduce the growth rate of Mtb. Whilst these experiments show the efficacy of the drugs against POPULATIONS of cells which are tolerant to other anti-TB drugs the experiments do not measure killing of the specialised sub-population of persister cells. To do this the authors would need to perform kill curves to test for changes in the biphasic kill curve (see more about this below). This just need rephrasing so that it doesn't mislead the reader. This also needs to be considered throughout the rest of the document.

We agree that the term "persisters" may not be accurate in this context. We have replaced the term "persisters" by "antibiotic-tolerant, non-replicating mycobacteria" or "non-replicating mycobacteria" throughout the text.

In the abstract the following sentence was amended (Lines 12-14):

"The drug combination was bactericidal against replicating and antibiotic-tolerant, non-replicating mycobacteria, and increased efficacy relative to that of a single drug in a mouse model."

In the Introduction section, the following sentences were revised (Lines 50-54):

"Hence, with the aim of shortening TB treatment, it is only logical to pursue the development of novel drugs that are active against antibiotic tolerant, non-replicating M. tuberculosis. Although the physiology of non-replicating mycobacteria is not fully understood, bioenergetics is a validated target space (Berube et al, 2019; Gengenbacher et al, 2010; Koul et al, 2008; Rao et al, 2008)."

In the Results section, the following sentence was revised (Lines 236-238):

"The main limitation of Q203 and related QcrB inhibitors lies in their lack of bactericidal efficacy (Foo et al, 2018; Kalia et al., 2017)."

In the Discussion section, the following sentence was revised (Lines 272-276):

"We and others have established that oxidative phosphorylation is essential in the maintenance of bioenergetics homeostasis in antibiotic-tolerant hypoxic (Rao et al., 2008) and nutrient-starved (Gengenbacher et al., 2010) M. tuberculosis, opening a scientific rationale to eradicate antibiotic-tolerant, non-replicating subpopulations (Cook et al, 2014; Hards & Cook, 2018; Kalia et al., 2017; Koul et al., 2008)."

(2) It is interesting that the drug combination was more effective against BCG as compared with Mtb. Why do the authors think this is the case? It seems worthy of a line of discussion. And considering this its strange that for the identified resistant strain the authors only investigated BCG. Were the Mtb drug resistant strains also investigated?

The *M. bovis* BCG Cyt-*bcc:aa₃* and Cyt-*bd* share 100% sequence similarity compared to the *M. tuberculosis* H37Rv counterparts (Brosch *et al.*, 2007; Lew *et al.*, 2011; Data ref: Garnier T., 2006; Data ref: Lew J. M., 2012). The reason behind the differences in the effectiveness of the drug combination is unclear but could possibly be due to differences in the expression levels of the terminal oxidases in laboratory-adapted *M. tuberculosis* strains, as described before (Arora *et al.*, 2014).

The *M. tuberculosis* escape mutants were not investigated because we have lost access to the BSL-3 laboratory in which the mutants were stored. Due to the COVID-19 pandemic, the laboratory is dedicated to SARS-CoV2-related work only for the foreseeable future.

The following sentence was added to the main text (lines 212-214):

*“Given that the Cyt-*bcc:aa₃* and Cyt-*bd* of *M. bovis* BCG and *M. tuberculosis* H37Rv share 100% sequence similarity, escape mutants isolated from *M. bovis* BCG were selected as representatives in the subsequent studies.”*

(3) For some of the experiments there needs to be some justification on the strain use as data seems to be presented from BCG, Mtb and *M. smegmatis* in Figure 1.

Referee 2 had a similar concern that was addressed (please see the response to the first major comment of referee #2).

(4) For the experiments where the authors measure the killing efficiency of the drug combinations in the Betts starvation model and a hypoxic model the authors pick one time point. Although there is an inoculum bar there doesn't seem to be a drug free control on figure 4b and 4c whereas there is a control on the growing cells (4a) and there is no Isoniazid control on figure 4a which are important and should be added. Im convinced that the drug combination works on these slow growing drug tolerant cells but without a time course the authors cannot comment on the effect on the persister population.

Figures 4B and C were modified to include the drug free controls.

We usually do not use isoniazid as a positive control in kill-kinetic experiments against replicating mycobacteria because of the rapid emergence of resistance *in vitro* (Gumbo *et al.*, 2007; Vilchèze *et al.*, 2018). Isoniazid kills mycobacteria rapidly *in vitro*, usually achieving 3-4 log₁₀ of killing after 2-4 days of treatment. However, the rapid emergence of resistance to isoniazid result in a resurgence of multiplication from day 5-6 onward (Gumbo *et al.*, 2007; Vilchèze *et al.*, 2018). Given the delayed bactericidal effect of bedaquiline and Q203 (against Δ *cydAB* strain) (Kalia *et al.*, 2017), we estimated that isoniazid is not an appropriate control drug to be used to compare the bactericidal potency of ND-011992+Q203 against replicating mycobacteria in a 10-day assay. Instead, we used bedaquiline, which is known to be bactericidal against replicating and non-replicating mycobacteria and acting on the same pathway as ND-011992+Q203.

References

- Arora K, Ochoa-Montaña B, Tsang PS, Blundell TL, Dawes SS, Mizrahi V, Bayliss T, Mackenzie CJ, Cleghorn LAT, Ray PC *et al* (2014) Respiratory flexibility in response to inhibition of cytochrome C oxidase in *Mycobacterium tuberculosis*. *Antimicrobial agents and chemotherapy* 58: 6962-6965
- Beites T, O'Brien K, Tiwari D, Engelhart CA, Walters S, Andrews J, Yang H-J, Sutphen ML, Weiner DM, Dayao EK *et al* (2019) Plasticity of the *Mycobacterium tuberculosis* respiratory chain and its impact on tuberculosis drug development. *Nature Communications* 10: 4970
- Berube BJ, Russell D, Castro L, Choi S-r, Narayanasamy P, Parish T (2019) Novel MenA Inhibitors Are Bactericidal against *Mycobacterium tuberculosis* and Synergize with Electron Transport Chain Inhibitors. *Antimicrobial Agents and Chemotherapy* 63: e02661-02618
- Brosch R, Gordon SV, Garnier T, Eiglmeier K, Frigui W, Valenti P, Dos Santos S, Duthoy S, Lacroix C, Garcia-Pelayo C *et al* (2007) Genome plasticity of BCG and impact on vaccine efficacy. *Proc Natl Acad Sci U S A* 104: 5596-5601
- Cook GM, Hards K, Vilcheze C, Hartman T, Berney M (2014) Energetics of Respiration and Oxidative Phosphorylation in Mycobacteria. *Microbiology spectrum* 2
- Foo CS, Lupien A, Kienle M, Vocat A, Benjak A, Sommer R, Lamprecht DA, Steyn AJC, Pethe K, Piton J *et al* (2018) Arylvinylpiperazine Amides, a New Class of Potent Inhibitors Targeting QcrB of *Mycobacterium tuberculosis*. *mBio* 9: e01276-01218
- Garnier T. (2006) GenBank AM408590.1 (<https://www.ncbi.nlm.nih.gov/nuccore/AM408590>). [DATASET]
- Gengenbacher M, Rao SP, Pethe K, Dick T (2010) Nutrient-starved, non-replicating *Mycobacterium tuberculosis* requires respiration, ATP synthase and isocitrate lyase for maintenance of ATP homeostasis and viability. *Microbiology* 156: 81-87
- Gumbo T, Louie A, Liu W, Ambrose Paul G, Bhavnani Sujata M, Brown D, Drusano George L (2007) Isoniazid's Bactericidal Activity Ceases because of the Emergence of Resistance, Not Depletion of *Mycobacterium tuberculosis* in the Log Phase of Growth. *The Journal of Infectious Diseases* 195: 194-201
- Hards K, Cook GM (2018) Targeting bacterial energetics to produce new antimicrobials. *Drug resistance updates : reviews and commentaries in antimicrobial and anticancer chemotherapy* 36: 1-12
- Hards K, Robson JR, Berney M, Shaw L, Bald D, Koul A, Andries K, Cook GM (2015) Bactericidal mode of action of bedaquiline. *Journal of Antimicrobial Chemotherapy* 70: 2028-2037
- Heikal A, Hards K, Cheung C-Y, Menorca A, Timmer MSM, Stocker BL, Cook GM (2016) Activation of type II NADH dehydrogenase by quinolinequinones mediates antitubercular cell death. *Journal of Antimicrobial Chemotherapy* 71: 2840-2847
- Kalia NP, Hasenoehrl EJ, Ab Rahman NB, Koh VH, Ang MLT, Sajorda DR, Hards K, Grüber G, Alonso S, Cook GM *et al* (2017) Exploiting the synthetic lethality between terminal respiratory oxidases to kill *Mycobacterium tuberculosis* and clear host infection. *Proc Natl Acad Sci U S A* 114: 7426-7431
- Kalia NP, Lee BS, Ab Rahman NB, Moraski GC, Miller MJ, Pethe K (2019) Carbon metabolism modulates the efficacy of drugs targeting the cytochrome bc₁:aa₃ in *Mycobacterium tuberculosis*. *Scientific Reports* 9: 8608
- Koul A, Vranckx L, Dendouga N, Balemans W, Van den Wyngaert I, Vergauwen K, Göhlmann HWH, Willebrords R, Poncelet A, Guillemont J *et al* (2008) Diarylquinolines Are Bactericidal for Dormant Mycobacteria as a Result of Disturbed ATP Homeostasis. *Journal of Biological Chemistry* 283: 25273-25280

Lamprecht DA, Finin PM, Rahman MA, Cumming BM, Russell SL, Jonnala SR, Adamson JH, Steyn AJC (2016) Turning the respiratory flexibility of *Mycobacterium tuberculosis* against itself. *Nature Communications* 7: 12393

Lew JM, Kapopoulou A, Jones LM, Cole ST (2011) TubercuList--10 years after. *Tuberculosis (Edinburgh, Scotland)* 91: 1-7

Lew J.M. (2012) RefSeq NC_000962.3 (https://www.ncbi.nlm.nih.gov/nucore/NC_000962.3). [DATASET]

Lu P, Heineke MH, Koul A, Andries K, Cook GM, Lill H, van Spanning R, Bald D (2015) The cytochrome bd-type quinol oxidase is important for survival of *Mycobacterium smegmatis* under peroxide and antibiotic-induced stress. *Scientific Reports* 5: 10333

Moraski GC, Markley LD, Chang M, Cho S, Franzblau SG, Hwang CH, Boshoff H, Miller MJ (2012) Generation and exploration of new classes of antitubercular agents: The optimization of oxazolines, oxazoles, thiazolines, thiazoles to imidazo[1,2-a]pyridines and isomeric 5,6-fused scaffolds. *Bioorganic & medicinal chemistry* 20: 2214-2220

Moraski GC, Oliver AG, Markley LD, Cho S, Franzblau SG, Miller MJ (2014) Scaffold-switching: an exploration of 5,6-fused bicyclic heteroaromatics systems to afford antituberculosis activity akin to the imidazo[1,2-a]pyridine-3-carboxylates. *Bioorg Med Chem Lett* 24: 3493-3498

Pecsi I, Hards K, Ekanayaka N, Berney M, Hartman T, Jacobs WR, Jr., Cook GM (2014) Essentiality of succinate dehydrogenase in *Mycobacterium smegmatis* and its role in the generation of the membrane potential under hypoxia. *mBio* 5

Rao SP, Alonso S, Rand L, Dick T, Pethe K (2008) The protonmotive force is required for maintaining ATP homeostasis and viability of hypoxic, nonreplicating *Mycobacterium tuberculosis*. *Proc Natl Acad Sci U S A* 105: 11945-11950

Rybniker J, Vocat A, Sala C, Busso P, Pojer F, Benjak A, Cole ST (2015) Lansoprazole is an antituberculous prodrug targeting cytochrome bc(1). *Nature Communications* 6: 7659

Tiwari R, Moraski GC, Krchňák V, Miller PA, Colon-Martinez M, Herrero E, Oliver AG, Miller MJ (2013) Thiolates Chemically Induce Redox Activation of BTZ043 and Related Potent Nitroaromatic Anti-Tuberculosis Agents. *Journal of the American Chemical Society* 135: 3539-3549

Vilchère C, Copeland J, Keiser TL, Weisbrod T, Washington J, Jain P, Malek A, Weinrick B, Jacobs WR (2018) Rational Design of Biosafety Level 2-Approved, Multidrug-Resistant Strains of *Mycobacterium tuberculosis* through Nutrient Auxotrophy. *mBio* 9: e00938-00918

20th Oct 2020

Dear Dr. Pethe,

Thank you for the submission of your revised manuscript to EMBO Molecular Medicine. I am pleased to inform you that we will be able to accept your manuscript pending the following final amendments:

***** Reviewer's comments *****

Referee #3 (Comments on Novelty/Model System for Author):

The authors have used a variety of appropriate in vitro and in vivo models

Referee #3 (Remarks for Author):

The authors have satisfactorily addressed all of the reviewers comments with their modified text and additional experiments.

The authors performed the requested editorial changes.

The authors performed the requested changes.

Corresponding Author Name:	Kevin Pethe
Journal Submitted to:	EMBO Molecular Medicine
Manuscript Number:	EMM-2020-13207